



**Evaluation of biomass burning aerosols in the HadGEM3**
**climate model with observations from the SAMBBA field**
**campaign**
B. T. Johnson[1], J. M. Haywood[1,2], J. M. Langridge[1], E. Darbyshire[3], W. T. Morgan[3], K.
Szpek[1], J. Brooke[1], F. Marenco[1], H. Coe[3], P. Artaxo[4], K. M. Longo[5], J. Mulcahy[1], G. Mann[6],
M. Dalvi[1], and N. Bellouin[7]
[1]Met Office, Exeter, UK
[2]CEMPS, University of Exeter, Exeter, UK
[3]Centre for Atmospheric Science, University of Manchester, Manchester, UK
[4]Physics Institute, University of São Paulo, São Paulo, Brazil
[5]National Institute for Space Research (INPE), São José dos Campos, Brazil
[6]National Centre for Atmospheric Science, School of Earth and Environment, University of
Leeds, Leeds, UK
[7]Department of Meteorology, University of Reading, Reading, UK
Correspondence email: ben.johnson@metoffice.gov.uk





## 1   Abstract

We present observations of biomass burning aerosol from the South American Biomass
Burning Analysis (SAMBBA) and other measurement campaigns, and use these to evaluate
the representation of biomass burning aerosol properties and processes in a state-of-the-art
climate model. The evaluation includes detailed comparisons with aircraft and ground data,
along with remote sensing observations from MODIS and AERONET. We demonstrate
several improvements to aerosol properties following the implementation of the GLOMAP-
mode modal aerosol scheme in the HadGEM3 climate model. This predicts the particle size
distribution, composition and optical properties, giving increased accuracy in the
representation of aerosol properties and physical-chemical processes over the CLASSIC bulk
aerosol scheme previously used in HadGEM2. Although both models give similar regional
distributions of carbonaceous aerosol mass and Aerosol Optical Depth (AOD), GLOMAP-
mode is better able to capture the observed size distribution, single scattering albedo, and
Ångström exponent across different tropical biomass burning source regions. Both aerosol
schemes overestimate the uptake of water compared to recent observations, CLASSIC more
so than GLOMAP-mode, leading to a likely overestimation of aerosol scattering, AOD and
single scattering albedo at high relative humidity. Observed aerosol vertical distributions were
well captured when biomass burning aerosol emissions were injected uniformly from the
surface to 3km. Finally, good agreement between observed and modelled AOD was gained
only after scaling up GFED3 emissions by a factor of 1.6 for CLASSIC and 2.0 for
GLOMAP-mode. We attribute this difference in scaling factor mainly to different
assumptions for the growth of aerosol mass during ageing via oxidation and condensation of
organics.

## 25   1   Introduction

Biomass burning is a major source of tropospheric aerosol globally (van der Werf et al., 2010)
and dominates the aerosol burden in many tropical regions. Carbonaceous aerosols are
produced from open burning of vegetation, including both wild fires and managed fires for
clearing forest, pasture and arable land. These aerosols have a wide range of impacts
(Voulgarakis and Field, 2015) including short-term influences on local and regional weather
(e.g. Kolusu et al., 2015) and significant impacts on regional air quality and human health
(Johnston et al., 2012; Reddington et al. 2015). They also have a significant role in climate





change as they affect the global energy budget in a number of ways (e.g. IPCC, 2013; Bauer
and Menon 2012, Haywood and Boucher 2000).
The aerosols emitted from Biomass Burning (BB) are composed primarily of organic carbon
and black carbon and they both scatter and absorb solar radiation in the atmosphere. Such
"aerosol-radiation interactions" lead to large reductions of surface insolation and significant
radiative heating of the atmosphere (Ramanathan and Carmichael, 2008; Johnson et al.,
2008a; Malavelle et al., 2011; Milton et al., 2008). These effects can suppress the
hydrological cycle and influence regional atmospheric circulation affecting global
precipitation patterns (Wu et al., 2013; Ramanathan et al., 2001). The enhancement of
particulate numbers by BB can also increase the concentration of cloud condensation nuclei
modifying cloud microphysical properties (Spracklen et al., 2011). This can brighten clouds
(Twomey 1974) and modelling studies have also shown that smoke (aerosol) from BB can
delay the onset of precipitation and influence the evolution of convective clouds (Andreae et
al., 2004; Feingold et al. 2001). The localised heating associated with absorption of solar
radiation by the emitted particles can also suppress convection and change regional cloud
cover via the semi-direct aerosol effect (Koren et al., 2008; Tosca et al., 2014).
Quantifying the impact of BB aerosol emissions on the global radiation budget and climate is
therefore difficult with many competing effects and sources of uncertainty (Ten Hoeve et al.,
2012; Ward et al. 2012). Recent assessments suggest that on a global basis, changes in the top
of the atmosphere (TOA) radiation budget resulting from increased scattering due to aerosol
emitted from BB is approximately cancelled by increased absorption by the aerosol (Myhre et
al., 2013; Shindell et al., 2013; Bellouin et al., 2013). However, the extent to which scattering
and absorption compensate varies regionally as it depends on many factors including the
surface albedo, cloud cover, and the optical properties of the aerosol. In particular, the single
scattering albedo (e.g. Myhre et al., 2008), and the vertical distribution of the absorbing
aerosol relative to clouds (e.g. Samset et al., 2013) have a strong influence on this potential
balance. Absorption depends mainly on the black carbon content of the aerosol, but a
significant contribution in the UV, and to a lesser extent visible spectrum, can come from
organics, i.e. brown-carbon (Saleh et al., 2014).
Overall, BB aerosol emissions are estimated to lead to a global mean negative Effective
Radiative Forcing (ERF) as aerosol-cloud interactions in models are shown to exert a negative
forcing that outweighs any small positive forcing from aerosol-radiation interactions. This is





expected to have a cooling influence on global climate but the ERF and global temperature
responses are estimated to be relatively small, compared to those from sulphates or black
carbon from fossil fuel combustion (Jones et al., 2007; Shindell et al., 2013). Nevertheless,
increases in aerosol due to BB have potentially important impacts on regional climates, via
changes in atmospheric circulation and shifts in precipitation (Tosca et al., 2010, 2013; Ott et
al., 2010; Zhang et al., 2009; Jones et al., 2007).
Recent studies have also highlighted more complex Earth-system interactions associated with
BB emissions. By scattering solar radiation and increasing the ratio of diffuse to direct
radiation at the surface, aerosol can enhance photosynthesis over tropical forests, increasing
Net Primary Productivity (NPP) and carbon uptake (Rap et al., 2015; Mercado et al. 2009).
On the other hand, tropospheric ozone produced due to NOx emissions from fires can damage
plants, reducing NPP (Pacifico et al. 2015). Mao et al. (2013) also showed that emission of
aerosol and trace gases from BB led to increases in global tropospheric ozone and methane
lifetime. In their study this led to a positive radiative forcing that offset the negative radiative
forcing from the sum of aerosol-radiation and aerosol-cloud interaction effects.
Quantifying these wide-ranging impacts of BB on climate, air quality and the earth-system
relies on the accurate representation of BB processes and aerosol properties in global models.
It is therefore important to evaluate their simulation in models with observations to reduce
inherent biases and identify priorities for future improvements to emissions, processes and
techniques used to represent aerosol properties. The properties of aerosols in BB-dominated
air masses have been investigated during a number of field experiments (e.g. Kaufmann et al.,
1998; Swap et al., 2002; Haywood et al., 2008), and reviewed by Reid et al. (2005a, 2005b)
and Martin et al. (2010). A new set of observations is now available from the South American
Biomass Burning Analysis (SAMBBA), a field campaign that took place in Brazil during 14
September – 4 October 2012. The measurement campaign was a joint UK-Brazil project led
by the Met Office and NERC, in collaboration with the National Institute of Space Studies in
Brazil (INPE) and University of Sao Paulo (USP) in Brazil. The campaign involved the UK
Facility for Airborne Atmospheric Measurements (FAAM) BAe-146 atmospheric research
aircraft coordinated with a range of ground-based observations (Allan et al., 2014; Brito et al.,
2014; Marenco et al., 2016). The airborne campaign comprised 20 flights investigating
aerosol properties, atmospheric chemistry, clouds, meteorology and the radiation budget over





Amazonia. The flights provided intensive measurement of aerosols across Amazonia
including aerosol dominated by BB emissions.
In this study we combined the observations from SAMBBA with those from previous
campaigns and from long-term remote sensing observations (MODIS, AERONET) to
evaluate the representation of Biomass Burning Aerosol (BBA) in a state-of-the-art global
climate model, the Hadley Centre Global Environment Model version 3 (HadGEM3). We
evaluate two aerosol schemes: (i) the mass-based CLASSIC aerosol scheme, (ii) the
microphysical scheme GLOMAP-mode. CLASSIC was previously used in HadGEM2-ES for
CMIP5 (Bellouin et al., 2011) and in this study is used within HadGEM3. GLOMAP-mode
has been implemented more recently in the Met Office Unified Model and is available in
HadGEM3 (e.g. Bellouin et al., 2013) (in some publications configurations of HadGEM3 that
include GLOMAP-mode have been referred to as HadGEM-UKCA). The study focuses on
aerosol properties important in simulating aerosol-radiation interactions, including the global
distribution of aerosol and their physical, chemical and optical properties. The study provides
an assessment of the influence of biomass burning on aerosol properties, as simulated by each
scheme, and assesses some of the assumptions commonly used to represent BB aerosol
emissions and aerosol processes in global models.

## 2  Methods

### 2.1  HadGEM3 model configuration

This work uses global simulations of the Met Office Unified Model (MetUM) within the
framework HadGEM3 (Hewitt et al., 2011). The scientific configuration of the physical
model was from the Global Atmosphere 7 (GA7) configuration and our simulations ran with a
resolution of N96 (1.25° x 1.875°) and 85 vertical levels. Sea surface temperatures and sea ice
were prescribed using reanalysed daily-varying fields for the period 2002 – 2011 based on the
methodology of Reynolds et al. (2007) (as used in Atmosphere Model Intercomparison
Project). The atmospheric circulation was free-running including aerosol-radiative effects
from either CLASSIC or GLOMAP-mode. The atmospheric physics configuration includes
some updates to atmospheric processes over previous configurations presented in Williams et
al. (2015) and Walters et al. (2014). The main update affecting this study is the
implementation of the Global Model for Aerosol Processes (GLOMAP-mode) (Mann et al.,





2010) modal aerosol scheme. The implementation of GLOMAP-mode in the MetUM took
place as part of the UKCA (United Kingdom Chemistry and Aerosol) project along with
several alternative atmospheric chemistry schemes. In this study we use an offline-chemistry
configuration where concentrations of gas phase chemical species [ozone ($O_3$), hydrogen
peroxide ($H_2O_2$), and the hydroxyl (OH), nitrate ($NO_3$) and hydroperoxyl ($HO_2$) radicals]
required for the oxidation of aerosol precursor species are provided as monthly mean
climatologies. The climatology of oxidants was generated from a previous 20-year simulation
that included on-line gas-phase atmospheric chemistry using the UKCA combined
tropospheric and stratospheric chemistry scheme (O'Connor et al., 2014; Morgenstern et al.,
2009). For this study a parallel simulation was also run with the same model configuration
except that aerosols were simulated by the CLASSIC (Coupled Large-scale Aerosol Scheme
for Simulations in Climate Models) aerosol scheme. CLASSIC was the aerosol scheme used
in HadGEM2, including Hadley Centre contributions to the fifth Coupled Model
Intercomparison Project (CMIP5) (Bellouin et al., 2011). As in the GLOMAP-mode
simulation a climatology of oxidants was used, generated from previous simulations with the
UKCA atmospheric chemistry schemes. For both aerosol schemes fire emissions of BBA
were taken from the Global Fire Emission dataset (GFED) version 3.1 (van der Werf et al.,
2010). Details of how these were implemented are given in section 2.3. Anthropogenic
emissions of $SO_2$ and carbonaceous aerosol (from fossil fuel and bio-fuel) for both aerosol
schemes were based on the 10-year average emissions from 2002-2011. These data were
provided by MACC/CityZEN (via ECCAD-Ether at http://eccad.sedoo.fr) that interpolates
across this time frame using historical emissions for 2000 from ACCMIP (Lamarque et al.,
2010) and emissions for 2005 and 2010 from the RCP8.5 scenario (Granier et al. 2011; Diehl
et al., 2012). Volcanic degassing emissions of $SO_2$ were taken from Andres and Kasgnoc
(1998). Emissions of di-methyl sulphide (DMS) were calculated from the Kettle et al. (1999)
ocean DMS climatology with the Liss and Merlivat (1986) surface-exchange
parameterization. Stratospheric aerosol was represented via the climatology from Cusack et
al. (1998). Nitrate aerosols were not included in this study.





## 2.2 Representation of aerosols

### 2.2.1 CLASSIC

CLASSIC is a mass-based or "bulk" aerosol scheme that represents a range of aerosol species (sulphate, fossil-fuel soot, fossil-fuel organic carbon, BBA, sea salt, and mineral dust) as separate externally-mixed species with specified physical and optical properties. A full description of the scheme is available in the appendix of Bellouin et al. (2011). CLASSIC includes a representation of the sulphur cycle for the gas-phase and aqueous-phase production of sulphate aerosol. Carbonaceous aerosols are represented as three separate species depending on their emission source (soot, fossil-fuel organic carbon, BBA). Each has different assumptions regarding their physical, chemical and optical properties. The representation of the BBA species is based on the aircraft observations of Haywood et al. (2003) and Abel et al. (2003) obtained over Southern Africa during SAFARI-2000 and is described in more detail below. Mineral dust is simulated by the 6-bin scheme of Woodward (2001), with modifications in Woodward (2011). CLASSIC uses a diagnostic scheme for wind-driven sea salt, i.e. assumes no transport of sea salt aerosol over land. Secondary Organic Aerosol (SOA) is not modelled explicitly by CLASSIC but the contribution to AOD and radiative effects is included using an offline climatology. The SOA climatology is provided by the UK Met Office Chemistry Transport Model (STOCHEM) (Derwent et al. 2003) based on the emission of isoprene from biogenic sources.

The BBA species includes a fresh mode to represent the primary particles, and an aged mode to represent the aerosols after chemical ageing and growth. A third tracer is used to track the mass of in-cloud BBA particles that are either lost via wet deposition or return to the aged BBA mode as rain water is lost via evaporation. The size distribution for each mode is represented by a single log-normal with a standard deviation of 1.3 and mean diameter of 0.2µm for the fresh mode and 0.24µm for the aged mode. These are assumed to be an internal mixture of Black Carbon (BC) and Organic Carbon (OC) with an organic carbon mass fraction of 91.5 % for the fresh mode and 94.6 % for the aged mode. The ageing process occurs on a 6-hour e-folding timescale and during the transfer from the fresh to aged mode the aerosol mass is increased by a factor of 1.62. This representation of aerosol ageing is based on the evolution of aerosol properties in a large smoke plume observed during SAFARI-2000 (Abel et al., 2003). Optical properties are calculated from Mie theory with the Refractive Index (RI) computed as the volume-weighted average of the BC and OC components





assuming an aerosol mass density of 1.35g/cm$^3$ for the OC and 1.7g/cm$^3$ for the BC. The RI
of the BC component is based on WCP (1983) (1.75 – 0.44i at 550nm) and the RI of the OC
component is assumed to be 1.53 – 0.0i across the solar spectrum. This gives an RI of 1.54 -
0.025i for the fresh mode and 1.54 – 0.018i for the aged mode, in the mid-visible (550nm).
Both species are hygroscopic with empirical growth curves from Magi and Hobbs (2003)
(Section 3.5).

### 2.2.2  GLOMAP-mode

The GLOMAP-mode (Global Model for Aerosol Processes) scheme (Mann et al., 2010) has
an entirely different modelling philosophy to CLASSIC, being an aerosol microphysics
scheme including a size-resolved representation of the key processes which alter the particle
physical and chemical properties during its lifecycle (e.g. Mann et al., 2014). The
configuration of GLOMAP-mode in this study (GA7) includes four soluble modes
(nucleation, Aitken, accumulation, coarse) and one insoluble Aitken mode, and includes the
components of sulphates, particulate organic matter, black carbon and sea salt. Aerosol
particles within any given mode are assumed to be an internal mixture of the chemical
constituents in that mode. Particles within a mode can grow by condensation and coagulation.
Aerosol mass and number can also be transferred from smaller to larger modes, either via
coagulation between the modes, or as the diameter of particles within a mode exceeds the
specified limit for that mode (Mann et al., 2010). Insoluble Aitken particles also age as
sulphuric acid and oxidised organic vapours condense onto them and the aerosol are
transferred to the accumulation soluble mode when the coating exceeds 10 mono-layers.
Although GLOMAP-mode generally treats mineral dust within the modal framework, in the
same way as other aerosol components, a modal representation for mineral dust was not
included in the GA7 configuration of the atmospheric model. Therefore, dust was modelled
separately using the CLASSIC dust scheme as in Bellouin et al. (2013). The simulation with
GLOMAP-mode included primary aerosol emissions from biomass burning, bio-fuel and
fossil fuel combustion sources, interactive sea spray emissions, and sub-grid sulphate particle
formation (so-called "primary sulphate"), assumed to be 2.5 % of emitted SO$_2$. The scheme
explicitly represents the secondary aerosol particle source from binary nucleation of sulphuric
acid vapour and water vapour following the nucleation rate of Kulmala et al. (1998).
Secondary organic aerosol (SOA) mass forms following the oxidation of emitted biogenic
Volatile Organic Compounds (bVOCs), and the SOA is added to existing particles. In these





simulations, the only SOA-producing bVOC is a lumped monoterpene species taking
emissions from Guenther et al., (1995). The monoterpene reacts with OH, $NO_3$ and $O_3$
assuming a 26 % molar yield to the particle phase.
Aerosol emissions from biomass burning are assumed to have an initial (emitted) size
distribution given by a log-normal with a mean diameter of 0.15µm and standard deviation of
1.59, as used by Stier et al. (2005) and consistent with the range of log-normal parameters
fitted to BB aerosol size distributions in Fig. C2 of Dentener et al. (2006). The ratio of BC to
OC varies interactively in GLOMAP-mode depending on mixing of these components from
the range of sources mentioned above. There is currently no representation for the
condensation of VOCs from BB onto the aerosol phase and so unlike in CLASSIC the growth
of aerosol mass from secondary aerosol formation during ageing occurs only via condensation
of sulphate and oxidised bVOC. The BC component in GLOMAP-mode is assumed to be
hydrophobic whereas the organic aerosol component is assumed to be hydrophobic only in
the insoluble mode(s), and hygroscopic in the soluble modes. Further details on the
hygroscopic growth are given in section 3.5. As in CLASSIC, aerosol optical properties are
calculated from Mie theory with RI computed by volume-weighted averages depending on the
mixture of components within any given mode. The RI of the BC component, as in
CLASSIC, is based on WCP (1983) and the OC component is assumed to be non-absorbing
with an RI of 1.5 – 0.0i across the solar spectrum. Aerosol mass density for BC and OC are
assumed to be 1.5g/cm$^3$.
**2.3    Biomass burning aerosol emissions and scaling factors**
**2.3.1    Global emission scaling factor**
Fire emissions of BBA were taken from the Global Fire Emission Dataset (GFED) version 3.1
(van der Werf et al., 2010). Preliminary simulations with GFED3.1 emissions led to large
underestimates in modelled aerosol mass and AOD over tropical BB regions. Therefore, we
apply global scaling factors of 1.6 for CLASSIC and 2.0 for GLOMAP-mode (Table 1) to
increase the total BB aerosol emissions to give good agreement between modelled and
observed mid-visible AOD (see section 3.1). Note, scaling BB aerosol emissions to observed
AODs is explicit in top-down emission estimation methods such as the Quick Fire Emission
Dataset QFED (Darmenov and da Silva 2013) and the Fire Energetics and Emissions
Research (FEER) (Ichoku and Elison 2014) and these lead to global total particulate matter





emissions approximately 2-3 times greater than GFED3.1 (Ichoku and Elison 2014). Other
modelling studies have also found it necessary to apply global scaling factors to increase
aerosol emissions from BB sources to gain realistic AOD and/or particulate mass
concentrations (Kaiser et al., 2012; Marlier et al., 2013; Petrenko et al., 2012; Tosca et al.,
2013; Archer-Nicholls et al., 2016; Kolusu et al., 2015; Reddington et al., 2016). However,
we acknowledge that the discrepancy between modelled and observed AOD (prior to
emission scaling) could be due to other biases or missing processes in the models.
In the CLASSIC simulation the global scaling factor of 1.6 (Table 1) was applied to the total
mass emitted into the BBA tracer. For GLOMAP-mode a factor of 2.0 was applied to the BB
emissions of aerosol mass (both the OC and BC component) and number. The scaling used
here differs between the two aerosol schemes, and aims of doing so is to quantify the
magnitude of the discrepancy between modelled and observed AOD (prior to scaling), and
highlight the fact that the discrepancy depends on assumptions and processes internal to the
aerosol schemes themselves.
**2.3.2   Scaling of organic carbon to primary organic matter**
In this study emissions of the organic aerosol component are derived from the OC flux
provided by GFED3.1. As OC represents the mass of the carbon only, the contribution of
other elements (principally oxygen) to the total organic aerosol mass (i.e. Primary Organic
Matter POM) must be considered separately. In CLASSIC no scaling is applied to convert the
mass of OC to POM. In GLOMAP-mode OC is converted to POM assuming a POM:OC mass
ratio of 1.4 (Table 1). This conversion factor of 1.4 has been broadly used in atmospheric
models and was originally based on analysis of filter measurements of fresh urban emissions
from the 1970's onwards (see Turpin and Lim, 2001 and references therein). More recent
analyses of aerosol mass spectra (e.g. Aitken et al., 2008; Ng et al., 2010; Tiitta et al., 2014;
Brito et al., 2014) and preliminary analysis of airborne data from SAMBBA indicate
POM:OC ratios in the range 1.5 – 1.8 for fresh particles / near-source emissions from biomass
burning. Therefore, an upward adjustment from the 1.4 conversion factor widely assumed
may be warranted to more accurately simulate the aerosol mass emissions from BB. However,
the observations indicate considerable variability with aerosol age and source region with
POM:OC ratios increasing to 2.0-2.3 for aged and more highly oxidised aerosol. This
introduces considerable uncertainty in gauging a representative POM:OC for global models
where near-source ageing may not be represented.





### 2.3.3  Growth of organic aerosol component during ageing

In CLASSIC the condensation of VOCs onto BBA is represented in a simplified manner increasing the aerosol mass by a factor of 1.62 (Table 1) when the fresh BB mode is converted to the aged mode. This scaling factor is based on measurements from a large plume during SAFARI-2000 (Abel et al. 2003). However, the evidence for growth of aerosol mass in BB plumes is mixed. For example, Vakkari et al. (2014) concluded that oxidation and subsequent secondary aerosol formation were important in the evolution of smoke plumes 2-4 hours after emission. In contrast, other studies based on aircraft measurements of aerosol composition and emission ratios have shown no net mass gain, or even net loss of aerosol mass between fresh and aged plumes, despite oxidation (chemical aging). These studies include measurements from West Africa (Capes et al. 2008), from SAMBBA (Morgan et al. 2012), and from a synthesis of the West African measurements with three other campaigns (Jolleys et al., 2012). These suggest that evaporation of organic material after initial emission outweighs or at least compensates for mass added due to secondary formation of organic aerosol. The assumed growth in CLASSIC is therefore not fully supported by recent observational analyses and is an aspect of the scheme that must be considered as we evaluate the model.

The configuration of GLOMAP-mode here does not include secondary aerosol formation from VOCs emitted by biomass burning, or the associated variation of POM:OC during chemical ageing. This is acknowledged as a potentially large source of bias that may to some extent necessitate the global emission scaling.

### 2.3.4  Vertical injection height assumptions

Smoke plumes can rise several kilometres before detraining into the atmosphere, although this depends critically on fire size / heat flux and atmospheric stability (Freitas et al., 2007). Regional assessments show that the majority of smoke plumes detrain in the boundary layer with maximum plume heights typically below 2km, whereas vigorous plumes from some large fires can extend into the free troposphere up to altitudes 6 km or more in exceptional cases (Freitas et al., 2007; Kahn et al., 2008; Val Martin et al., 2010; Val Martin et al., 2012; Sofiev et al., 2012; Tosca et al., 2011). During SAMBBA the concentration of aerosol was generally highest in the lowest (2 – 3 km corresponding to the maximum height of the atmospheric boundary layer) and declined rapidly with height above this (Marenco et al., 2015; Darbyshire et al., in preparation, 2016a). Tenuous aerosol layers were frequently



observed in the mid-troposphere up to altitudes of 5 or 6 km but given the prevalence moist
convection during SAMBBA (mainly in the Western region) it was difficult to determine
whether these elevated layers related to plume injection heights or were the result of vertical
transport and detrainment from cumulus (in some cases pyrocumulus were also observed).
HadGEM3 does not include an explicit smoke plume-rise model but prescribes the vertical
profile of emissions depending on vegetation type provided by GFED. Following
recommendations from the 1st phase of AeroCom (see section 7 and Fig. 9 of Dentener et al.,
2006) fire emissions from peat fires, savannah and woodland are assumed to have small
plume rise and are emitted at the lowest model level, allowing sub-grid scale turbulence to
mix these through the boundary layer. Emissions from forest and tropical deforestation fires
are assumed to have more significant plume rise and are injected uniformly from the surface
to an assumed maximum injection height of 3km. These injection height assumptions were
used identically for both the CLASSIC and the GLOMAP-mode simulations to maintain
consistency.

## 2.4   Experimental design of simulations

Three simulations were completed, each with a 3 month spin up followed by a 10 year run
with emissions, SSTs and sea ice based on the years 2002-2011. These included simulations
with: (i) CLASSIC aerosols, (ii) GLOMAP-mode aerosols, (iii) GLOMAP-mode but with no
BB emissions. Simulation (iii) enables the contribution of BB emissions to AOD and aerosol
mass to be inferred from GLOMAP-mode. Apart from these changes in the simulation of
aerosols the scientific configuration of the atmospheric model was identical in all simulations.
In these simulations the atmospheric circulation was free-running (not nudged to
meteorological analyses) and so a ten year period is required to average over interannual
variability of meteorology. The selected time period 2002 – 2011 spans the last 10 years
where GFED3.1 data was available. One advantage of selecting this time period rather than
earlier years is that the GFED3.1 emissions benefit from inclusion of the burned area product
from Aqua MODIS from 2002 onwards. Unfortunately, the GFED3.1 data were not available
for 2012 (the period of the SAMBBA campaign). For this reason the evaluations against
aircraft campaign data in this study focus on the intrinsic properties of BBA (physical,
chemical and optical properties) that are expected to depend more on the vegetation and
burning practices in the observed regions than on year-to-year variability of burned area.

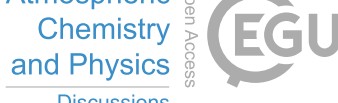

## 2.5   In situ observations from SAMBBA and other biomass burning campaigns

Aircraft measurements of aerosol properties have been taken from the SAMBBA campaign that took place in Brazil during Sept-Oct 2012. As the aerosol properties differed regionally (Darbyshire et al., in preparation, 2016b) we present average properties separately for the Western region (flights based from Rondonia: 7-12.5°S, 58-65°W) and Eastern region (flights over Tocantins: 10-12°S, 46.5-49°W) (Fig. 1). An overview of the flights and full details of instrumentation are provided in Darbyshire and Johnson (2012) and data processing methods will be described in Darbyshire et al. (in preparation, 2016b). Aircraft measurements have also been taken from the Dust and Biomass Burning Experiment (DABEX) over West Africa (7-15°N, 0-7°E) during Jan-Feb 2006 (Haywood et al., 2008; Johnson et al.; 2008) and from the Met Research Flight C130 aircraft during the Southern African Regional Science Initiative (SAFARI-2000) in Sept 2000 (Haywood et al. 2003) (15-25°S, 8-18°E). The boxes in Fig. 1 indicate the regions where the flights took place and where model data was averaged.

Common to each of the aircraft datasets is the use of a wing-mounted Passive Cavity Aerosol Spectrometer Probe (PCASP) to measure aerosol particle size distributions, a TSI three wavelength nephelometer (440, 550, 700nm) to measure aerosol scattering, and a single wavelength Particle Soot Absorption Photometer (523nm) to measure aerosol absorption and the SSA (when combined with the nephelometer). During SAMBBA the PCASP suffered some instrument / electronic processing errors after the first four flights (B731-734). Therefore, stringent quality checks on the data were employed to filter out affected data. After this, approximately 16 hours of PCASP data were available from 8 flights, with 75 % of this from the first four flights (B731-734) that focussed mainly on sampling aerosol dominated by biomass burning emissions. During SAMBBA, PCASP measurements of aerosol size distribution were supplemented by a GRIMM Optical Particle Counter (OPC), and a TSI Scanning Mobility Particle Sizer (SMPS). The aerosol composition was also measured during SAMBBA and DABEX. In both cases the sulphate mass and the Organic Aerosol (OA) (i.e. total carbonaceous aerosol mass from POM and Secondary Organic Aerosol) were measured by an Aerodyne Aerosol Mass Spectrometer (Capes et al., 2008). During SAMBBA the BC mass was estimated from a Single Particle Soot Photometer and during DABEX the BC mass was estimated from the PSAP assuming a mass absorption coefficient of $12m^2/g$. For each flight campaign the aircraft observations have been averaged over all available measurements taken in biomass burning conditions to provide campaign mean BB aerosol properties.



Ground-based observations of aerosol composition have also been used based on data
presented in Tiitta et al. (2014) from the Welgegund station in South Africa (Fig. 1). They
used an Aerosol Chemical Speciation Monitor to measure OA and sulphate and a Multi-Angle
Absorption Photometer to measure BC assuming a mass absorption coefficient of 6.6m$^2$/g.
We take an average composition from their measurements in September 2010.
**2.6   Remote sensing observations**
MODIS AOD retrievals have been obtained from the Aqua satellite. In this study we use
monthly mean level 3 MYD08_M3 data products and to aid the evaluation we include the
AOD products from both collection 5.1 and collection 6. In the case of collection 5.1 the dark
target (Levy et al., 2007, 2010) and ocean algorithms (Remer et al., 2005) have been used
where coverage is available, and the Deep Blue algorithm (Hsu et al., 2004, 2006) has been
used for pixels over bright land surfaces where dark-target retrievals were not available. For
collection 6 the merged product (Sayer et al., 2014) has been used that combines retrievals
from all three algorithms and includes various refinements to each (Sayer et al., 2013; Levy et
al., 2013). The monthly mean data has been averaged over the period 2003-2012 to create
long-term monthly means.
AERONET data have also been used for direct sun retrievals of AOD and for inversion
products of aerosol size distribution and optical properties. Six sites with strong biomass
burning influence were selected for use in this study: Alta Floresta (Brazil), Mongu (Zambia),
Ilorin (Nigeria), Chiang Mai (Thailand), Jaribu (N. Australia), and Bonanza Creek (Alaska)
(Fig. 1). We used monthly mean products from the version 2 algorithm (Dubovik and King,
2000; Dubovik et al., 2006) and used level 2 products in all cases except Chiang Mai where
level 1.5 data was used as level 2 data coverage was limited. Level 1.5 data is not fully cloud-
screen and calibrated so may not be as reliable. Long-term monthly mean averages were
calculated for 2002-2011.
**2.7 Averaging methods**
The aircraft in-situ observations presented in this study have been averaged over all available
measurements in biomass burning dominated conditions in each campaign or campaign sub-
region, to provide representative "campaign-mean" values. The data averaging methods for
SAMBBA will be described in more detail in Darbyshire et al. (in preparation, 2016b). The



DABEX campaign-means are taken from the observations of aged aerosol layers in (Johnson
et al., 2008). SAFARI-2000 campaign-means are based on a compilation of aged aerosol
measurements, as detailed in Haywood et al. (2003). Inevitably, aircraft flight patterns do not
provide unbiased spatial and temporal sampling of the atmosphere and tend to favour
sampling aerosol layers with medium-high aerosol loadings. However, by averaging over
large volumes of data focussed on regional sampling these aircraft datasets can provide useful
constraints on the physical, chemical and optical properties of the aged aerosol. Wherever
comparisons are made with model data, they are based on the 10-year (long-term) monthly
mean output from the models (September for SAMBBA and SAFARI-2000, January for
DABEX). For comparison with aircraft measurements, the model data has been averaged over
the latitude and longitude ranges of the relevant flight regions (boxes in Fig. 1) and over 0 – 5
km; the typical altitude range of the observed aerosol layers. For the comparison with
Welgegund surface measurements model data is taken from the lowest model level of the grid
box co-located with the site and for September, corresponding to the peak of the BB season in
Southern Africa. For comparisons with AERONET the 10-year (long-term) monthly mean
model output is selected for the gridbox co-located with the AERONET site and averaged
vertically to provide column-mean aerosol properties.
**3   Evaluation of CLASSIC and GLOMAP-mode with observations**
**3.1   Aerosol optical depth**
**3.1.1   Global AOD evaluation with MODIS**
Fig. 2 assesses the contribution of biomass burning to annual mean Aerosol Optical Depth
(AOD) at the global scale. For CLASSIC the contribution of BB emissions to the total AOD
(hereafter BBAOD) is straightforward as carbonaceous aerosol originating from biomass
burning emissions are represented as separate (externally mixed) species in the model. For
GLOMAP-mode aerosols from different sources are internally mixed and so BBAOD is
estimated as the difference in AOD between a simulation including BB emissions and one
without. The total AOD from the models and from MODIS collection 5 and 6 are also shown.
The results show that biomass burning dominates the annual mean AOD over S. America and
Central to Southern Africa, even though BB emissions are highly seasonal in these regions.
Biomass burning also makes strong contributions to annual mean AOD in parts of Indonesia,





South East Asia, and Northern Australia and to a less extent the Boreal forests of N. America and North East Asia. Globally BBA emissions account for 10 % of the total AOD in the CLASSIC simulation and 12 % in the GLOMAP-mode simulation. The spatial distributions of BBAODs are very similar in both models, which is not surprising since they are driven by the same physical model and emission dataset. The magnitude of BBAODs are also very similar, which again is not surprising as the emissions in each run were scaled separately to match the magnitude of the AOD observations presented here and in later figures. Some differences in BBAOD do occur between the two aerosol scheme due to differing representation of aerosol properties and processes. Overall BBAODs are slightly higher in GLOMAP-mode especially for the plume from central Africa.

A wider assessment of simulated AOD from GLOMAP-mode in HadGEM3 / GA7 is expected in a future study but we note from Fig. 2 that GLOMAP-mode has improved the distribution of AOD in several regions compared to simulation with CLASSIC. For instance, it has reduced the low bias over high latitude continents (as found previously in Bellouin et al., 2013) and reduced a high bias in the southern ocean associated with sea salt aerosol. The AOD over the Sahara and North African Atlantic coast and Arabian Peninsula appear too low in the simulations indicating that mineral dust emissions may have been too weak (in both cases simulated by CLASSIC). We note however that the GLOMAP-mode simulation also overestimates AOD in south-east Europe and eastern parts of USA which are dominated by anthropogenic sources of sulphate. A strong caveat in these comparisons is that the modelled AOD has not been sampled with the spatial and temporal incidence of the MODIS data. Schutgens et al. (2015) showed that this can result in considerable regional biases between modelled and observed monthly and annual mean AOD. In particular, the comparison may be of limited value at high latitudes (beyond $60^{\circ}$N or S) where retrievals are not possible for several months of the year (due to solar zenith angle being too high, or due to lack of solar illumination altogether). Some degree of sampling bias may occur in regions that are frequently overcast with cloud cover (e.g. marine stratocumulus regions including the South-East Atlantic). The modelled AOD has however been calculated based on the clear-sky relative humidity to avoid strong humidification biases in partially cloudy grid boxes.

### 3.1.2 Seasonal AOD in biomass burning regions with MODIS

Figs 3 and 4 focus on the contribution of biomass burning to AOD in the tropical regions. Fig. 3 shows the monthly mean BBAOD and AOD for September when BB emissions peak in the



Southern Hemisphere and equatorial regions. Fig. 4 shows the same for West African region
but for January when BB emissions peak in the zone 5 – 15° N. As in the global picture (Fig.
2) the simulations give very similar regional distributions of BBAOD and AOD. Overall the
modelled AOD in both simulations agrees very well with MODIS, especially over South
America and Indonesia. However, there are some discrepancies between modelled and
observed AODs in northern and southern parts of Africa. Firstly, the magnitude of AOD in
the plume over the South-East Atlantic is lower in the models than in MODIS (Fig. 3).  It is
not clear if this is due to poor model performance or biases related to limited temporal
sampling by the satellite over the marine stratocumulus region (personal communication
Andrew Sayer). MODIS collection 5 and collection 6 in particular, show a large contrast in
AOD between the plume over the ocean and the AOD over adjacent land areas of Southern
Africa. Secondly, in Fig. 4 the peak AOD and BBAOD in the models during January are
centred over central Africa (Congo basin) rather than over the Gulf of Guinea where MODIS
AOD peaks. Again, high cloud cover limits the spatial sampling over the Congo basin and
may affect the mean AOD retrieved from MODIS. This regional bias was noted in previous
modelling studies with GFED2 (Myhre et al., 2008; Johnson et al., 2008a) and may suggest
there is still an underestimation in West Africa (Liousse et al., 2012) and potentially an
overestimation of BB aerosol emissions in the Congo basin.  The comparison of modelled and
observed AOD over the BB regions of the Sahel (north of 10°) is less straightforward as
mineral dust aerosol contributes strongly to the total AOD.
**3.1.3  AOD comparison with AERONET**
To aid the evaluation of modelled AOD, six AERONET sites have been selected representing
locations that are strongly affected by seasonal biomass burning. Once again, due to the
tuning of total BB aerosol emissions in the simulations both CLASSIC and GLOMAP-mode
give very similar AOD and BBAOD at these locations during peak months (Fig. 5). The
seasonal cycle and peak AODs seem well captured at Alta Floresta (Amazonia) and Mongu
(Southern Africa). The comparison at Ilorin (West Africa) shows the model does not capture
the observed seasonal cycle of AOD. This again suggests an under-representation of BB
emissions across West Africa during Northern hemisphere winter. The secondary peak during
June-Sept, which is not shown in the AERONET observations, may be due to overestimation
of BB aerosol emissions from the Congo basin and long-range transport to West Africa.
BBAOD appears to be underestimated at Chiang Mai (South East Asia) and Jaribu (Northern



Australia), perhaps by a factor of 2, but slightly overestimated at Bonanza Creek (Alaska).
Whilst these results give clues as to where BB aerosol emissions may be over or under
estimated the differences between modelled and observed AOD may be affected by various
other sources of uncertainty in the models and measurements. In particular, temporal
sampling biases may affect the results (Schutgens et al., 2015) as we have not sampled the
model data to match AERONET retrieval times.

## 3.2   Aerosol composition

Fig. 6a and b show the column loading of fine-mode aerosol mass from the model simulations
across the tropical regions during September. This is the sum of Black Carbon (BC), Organic
Aerosol (OA) and sulphate (SU) from all anthropogenic and natural sources but excluding the
coarse-mode contribution from GLOMAP. Clearly the fine-mode aerosol is dominated by BB
sources over Africa, South America, Indonesia and Northern Australia. Figs 6 c–h show the
relative contributions of OA, BC and SU to this fine-mode mass. CLASSIC and GLOMAP-
mode give very similar spatial distributions for the modelled fine-mode aerosol mass loading
and composition. OA clearly dominates the fine-mode aerosol mass (Fig. 6c and d) in both
models across most of the region shown, where BB emissions dominate the aerosol loading.
The two exceptions are the northern edge of the domain and some stretches along the Pacific
coast of South America where sulphates dominate due to anthropogenic emissions of $SO_2$.   In
Fig 6 stipples mark grid columns where over 75 % of the fine-mode aerosol mass originates
from BBA emissions, based on the speciation in the CLASSIC simulation. These mark the
main BB plumes from S. America, Africa and Indonesia. In GLOMAP-mode where aerosols
internally mix the origin of the aerosol in a grid cell can not be traced to its emission source
but it seems reasonable to assume that the grid cells strongly influenced by BB emissions in
CLASSIC will also be strongly influenced by BB emissions in the GLOMAP-mode given that
the simulation are driven with the same emissions data and physical model configuration. The
similarity in the spatial distribution of BBAOD (Fig. 2 – 4) and aerosol composition (Fig. 6)
between the two models support this assumption. The same areas are therefore marked with
stipples in the GLOMAP-mode plots. The mean values beneath each plot indicate the mean
from the stippled areas.
In the main BB plumes (marked by stippling) the CLASSIC simulations show a slightly
higher mass fraction of OA and a slightly lower mass fraction of BC compared to GLOMAP-
mode with BC mass fraction averaging 5.1 % in CLASSIC and 7.2 % in GLOMAP-mode





(Fig. 6e and f). These differences are due to differences in the way that BB composition is
represented in the two schemes. In CLASSIC the ratio of BC to OA in the BBA species is
specified, whereas in GLOMAP-mode it varies depending on the BC and OC mass provided
by the emissions data, and the OC to POM ratio assumed in the model (currently 1.4). In
GLOMAP-mode Secondary Organic Aerosol (SOA) is also added interactively via the
oxidation and condensation of organic vapours from bVOCs. This decreases the BC mass
fraction in North Western Amazonia compared to South Eastern Amazonia and Southern
Africa. In CLASSIC bVOCs are not modelled explicitly but SOA has been included using a
biogenic aerosol climatology. This increases the OC mass, particularly over tropical forests,
and therefore leads to a lower BC mass fraction over tropical forests compared to Savannah
regions. The localized peak in BC mass fraction near to Lake Victoria in the CLASSIC
simulation is due to local anthropogenic BC emissions rather than BB emissions. This shows
up less in GLOMAP-mode as the regional loading of BC from BB sources is higher.
In situ measurements from three observation campaigns have been used to evaluate the
aerosol composition in the simulations. The observations include FAAM aircraft
measurements from Western Amazonia (Rondonia) and Eastern Amazonia (Tocantins) during
SAMBBA (Darbyshire et al., in preparation, 2016b), ground-based observations from the
Welgegund measurement station in South Africa (Vakkari et al. 2014), and FAAM aircraft
measurements from West Africa during DABEX (Capes et al. 2008). Fig. 7 compares the
observed and modelled aerosol composition by plotting the relative contributions from BC,
OA and sulphate to the total fine-mode aerosol. Nitrate, dust and sea salt have been excluded
from the analysis as nitrate was not available in the model simulations and accurate
measurements of dust and sea salt were not readily available from all observation campaigns.
Given that these components are neglected we can not provide a full analysis of the aerosol
composition here. The purpose of Fig 7 is rather to examine whether the relative proportions
of BC, OA and sulphate are in-line with the observational evidence (as these are the dominant
contributors to fine-mode mass and fine-mode AOD in the simulations).
In all cases the fine-mode aerosol is dominated by OA with modest contributions from
sulphate and generally a smaller contribution from BC. On the whole the models are able to
capture the typical make-up of the aerosol and some of the variations with region, such as the
higher contribution from sulphates in South Africa. GLOMAP-mode gives slightly higher BC
mass fractions than CLASSIC and in general GLOMAP-mode BC mass fractions are closer to





observed values. Modelling the BC mass fraction is of key importance for estimating
absorption and the sign of direct radiative forcing. GLOMAP-mode therefore shows some
improvement over CLASSIC, although it still appears to underestimate BC mass fraction
relative to the measurements from West Africa, Eastern Amazonia, and to a lesser extent in
South Africa. However, the use of the filter based absorption measurements in those datasets
may lead to a significant overestimation of observed BC mass (Lack et al., 2008). Also, note
that different mass absorption coefficients were assumed in the analyses of the DABEX
($12m^2/g$) and Welgegund ($6.6.m^2/g$) observations. Unifying this assumption to an
intermediate value of $10m^2/g$ would change the estimated BC mass fraction to 14.1 % for
DABEX and 8.2 % for Welgegund.
**3.3 Size distributions**
**3.3.1 Comparison with aircraft data**
Fig. 8 shows the size distributions from the models and in situ observations from the three
aircraft campaigns. The CLASSIC curve is simply the size distribution given by the average
mixture of fresh and aged BBA species in the model. Each of these CLASSIC modes is
represented by a single log-normal. Both modes have a small standard deviation of 1.3 and
the mean diameters are 0.2 µm for the fresh mode and 0.24µm for the aged mode. Combining
these gives a fairly narrow distribution peaking in the accumulation mode. The GLOMAP-
mode size distribution is the sum of all five modes (nucleation, Aitken soluble and insoluble,
accumulation soluble, coarse soluble). Each campaign includes data from a common PCASP
instrument but SAMBBA included a GRIMM OPC behind a low-turbulence inlet and a
Scanning Mobility Particle Sizer (SMPS). These instruments provide a dry aerosol size
distribution as heating tends to the remove water from the measured aerosol samples. The
three instruments from SAMBBA are in good agreement regarding the shape of the
accumulation mode and the rate of decline from the accumulation to coarse mode (0.3 –
1µm). To avoid mismatches from sampling different total concentrations, the PCASP size
distributions have been normalized to give a total concentration of unity, and other observed
and modelled curves have been normalized to match the peak amplitude of the PCASP.
The dry particle size distribution simulated by GLOMAP-mode is shown in Fig. 8 and
matches the observed size distributions remarkably well. The broad peak in aerosol number
around 0.2µm and the rate of decline either side of the peak seem well supported by the





available observations. The discrepancies between the GLOMAP-mode and observed size
distributions across the coarse mode (D > 1µm) are most likely because mineral dust is not
represented in this version of the modal scheme (this is certainly the reason in the DABEX
case; Fig. 8b). Another potential issue in the Amazon case is the absence in the model of any
representation of primary biological aerosol particles which may contribute significantly to
the observed coarse mode in this forested region (e.g. Scot et al., 2012), though such particles
are only likely to be important in the surface mixed layer. Also, measurements of low
concentrations of super-micron particles will have bigger uncertainties than measurements of
the accumulation–mode peaks. The agreement between GLOMAP-mode and the observations
across the accumulation mode (0.1 – 0.6µm) is partly due to a well chosen initial size
distribution that is assumed for primary emissions of BBA (this a log-normal with a mean
diameter of 0.15µm and standard deviation of 1.59 as used by Stier et al., 2005). This sets the
mass and number of particles emitted into the Aitken insoluble mode. Subsequently, as a
result of ageing these particles grow and are transferred to the accumulation soluble mode,
where most of the BC and OA mass ultimately resides. Here coagulation and condensation
create an internal mixture of sulphate, sea salt, OC, BC and water from all modelled sources.
The combination of a well chosen initial size distribution for the primary emissions, and
subsequent microphysical and chemical processes operating through the modal framework,
are therefore very successful in predicting the aerosol size distribution over BB regions.
CLASSIC provides a reasonable representation of the aerosol size distribution through the
centre of the accumulation mode (0.1 – 0.6µm) that is most important for optical properties in
the visible and near-infrared spectrum. CLASSIC naturally fits the SAFARI-2000 PCASP
observations (Fig. 8c), on which it was originally based (Haywood et al. 2003), but also fits
the DABEX and SAMBBA observations reasonably well across the intended size range.
It is interesting to note that the observed size distributions do not vary greatly across the
accumulation mode (0.1 – 0.6µm) between the three BB campaigns. These campaigns span
three of the main continental source regions of BBA (Fig. 9a) and include a range of biomes
and fire conditions. This finding of little variation in size distribution between different
biomass burning source regions suggests the approach of using a globally representative size
distribution in CLASSIC, and of using a single "emission size distribution" for all primary
biomass burning emissions in GLOMAP-mode is a reasonable approximation. We note
however that Dentener et al. (2006) present a synthesis of observations from a wider



collection of observations, suggesting considerable variation in size distribution (their Figs.
C1 and C2). These indicate apparently large changes in physical and optical properties
between different biomass burning source regions and/or following ageing of plumes. The
large differences shown in Dentener et al. (2006) could in part be related to differences in
systematic biases or sizing corrections applied to differences instruments, while here we
present coherent results from essentially the same instrument (Fig 9b).
**3.3.2  Comparison with AERONET size distributions**
In Fig. 10 AERONET retrievals of particle size distribution are used as an additional
constraint to assess the modelled aerosol size distribution. These are given in terms of particle
volume across the fine and coarse modes (0.1 – 15µm) and all distributions have been
normalized to give peak amplitudes of 1.  The overall shape of the distribution varies very
little from year to year (Fig 10a) with a dominant fine-mode peaking around 0.3µm. The
relative contribution from coarse-mode particles varies from year to year but is generally
small. A similar analysis was performed for Mongu and produced an almost identical fine-
mode size distribution giving some confidence that Alta Floresta is representative for tropical
biomass burning regions.
Fig. 10b compares the AERONET size distribution to the PCASP and GRIMM OPC aircraft
instrument data from the Western SAMBBA region. Again, all size distributions have been
normalized to give the same peak amplitude. It is encouraging that the PCASP gives an
almost identical size distribution to AERONET across the fine-mode. The GRIMM OPC size
distribution covers only a portion of the fine-mode size range but the data are consistent with
the existence of a peak at 0.3µm, a minimum around 1µm and a peak at coarser sizes. The
aircraft instruments do not agree so well with AERONET on the amplitude or diameter of the
coarse mode. The coarse-mode could be a mixture of mineral dust, primary biogenic particles
or fly ash from BB (Martin et al., 2010). Sampling issues may be a large source of error in the
PCASP and GRIMM measurements of super micron particles. However, the coarse-mode is
not the focus of the assessment here as the sources are unclear and it contributes very little (5
– 10 %) to the AOD or optical properties.
Fig. 10c compares the mean AERONET size distribution with the models. For CLASSIC the
size distribution of the BBA species is plotted whereas GLOMAP-mode is the column-mean
for September co-located with Alta Floresta. Both modelled size distributions peak at about
the same diameter (~0.3µm) as AERONET. The CLASSIC size distribution is a little





narrower than AERONET whereas GLOMAP-mode predicts about the same width as
AERONET. This increases confidence that GLOMAP-mode is able to predict aerosol size
distributions accurately, and is an improvement over the specified distribution in CLASSIC.

## 4  3.4  Optical properties

In this section the aerosol optical properties from the models are compared and evaluated
against AERONET retrievals and in-situ measurements from aircraft campaigns. The methods
for deriving optical properties are described below and results are then discussed separately
for each optical property.
Firstly the column-average moist aerosol properties have been calculated from the models to
assess how these vary regionally in the two aerosol schemes. The fine-mode specific
extinction coefficient ($k_{ext,fm}$) (Fig. 11a & b) was calculated as the ratio of fine-mode moist
AOD to fine-mode dry aerosol mass. In GLOMAP-mode the fine-mode includes the Aitken
soluble, Aitken insoluble and accumulation-soluble modes. In CLASSIC the fine-mode is
taken to include all sulphate and carbonaceous aerosol species. The Single Scattering Albedo
(SSA) (Fig.11c & d) has been calculated from the AOD and Absorption-AOD (AAOD) at
550nm, and the Ångström exponent (Å) (Fig. 11e & f) is calculated from the wavelength
dependence of AOD across 440 – 670nm. The stipples in Fig. 11 mark grid columns where
over 75 % of the fine-mode aerosol mass originates from BBA emissions (as in Fig. 6, based
on CLASSIC speciation) and the mean values beneath each plot indicate the mean from the
stippled areas.
Secondly, the modelled SSA and Å are compared for all months against AERONET retrievals
for Alta Floresta and Mongu (Fig. 12). Monthly mean SSA retrievals were not available in all
months of the year due to low temporal sampling frequency outside of the dry season
(inversions require AOD > 0.4 and cloud-free skies). In addition to AERONET level 2 criteria
we only accept a monthly mean if data were available from at least 3 separate days in that
month, and only calculate the long-term monthly mean if at least 3 monthly means were
available in the time series. The AERONET retrievals of Å relied on direct sun measurements
of AOD at 440 and 670nm and have better temporal sampling enabling long-term monthly
means to be calculated for every month.
Finally, Table 2 compares dry aerosol optical properties of SSA, Å, $k_{ext,fm}$, and asymmetry
parameter (g) from the models and from the mean values from the aircraft campaigns



(references provided in the table). The comparison is made for dry aerosol since heating tends
to dry the aerosol samples measured by the aircraft instruments. For CLASSIC, the optical
properties are specified and so values in Table 2 are simply derived by averaging together the
optical properties for fresh and aged BBA species, based on the typical mixture simulated
over the BB regions (10 % fresh, 90 % aged). For GLOMAP-mode the dry optical properties
in Table 2 were calculated from Mie theory using the dry size distribution and refractive
index for each of the fine modes (Aitken soluble, Aitken insoluble and accumulation-soluble)
and then averaged across the modes weighting by total extinction (or by scattering for g).

### 3.4.1  Fine-mode specific extinction coefficient (kext,fm)

The fine-mode moist specific extinction (Fig. 11a & b) varied quite widely in both models but
was generally higher in GLOMAP-mode, especially in areas where sulphates were more
dominant (see Fig. 6h). This is due to a high water uptake by sulphate in the current
GLOMAP-mode configuration. In the main BB plumes (marked by stipples), where OA
dominates the aerosol mass, the values of $k_{ext,fm}$ range from 5 – 10 m$^2$/g with the highest
values in both models over the moister regions of Indonesia and the lowest values in Southern
Africa where the average relative humidity was lower in the lower troposphere (not shown).
The average values from the BB plumes (stippled areas) are fairly similar with slightly lower
value of 6.2 m$^2$/g for CLASSIC and a value of 6.9m$^2$/g for GLOMAP-mode. Note these
values are indicative of the aerosol mixture as a whole rather and so are affected by the
representation of other aerosols also. In Table 2 the dry values of $k_{ext,fm}$ are similar for
CLASSIC (5.0 m$^2$/g) and GLOMAP-mode (4.5 – 4.8 m$^2$/g) and are within the range given by
the aircraft measurement campaigns (3.6 – 5.8 m$^2$/g). Note, in this case the dry value given for
CLASSIC corresponds to the BB species only.

### 3.4.2  Single scattering albedo (SSA)

The SSA of aerosol over BB dominated regions was generally lower in GLOMAP-mode than
in CLASSIC for both the ambient (moist) values (Fig. 11c and d; Fig. 12a and c) and dry
values (Table 2). This is consistent with the higher BC mass fraction in GLOMAP-mode (Fig
7). The lower dry SSA values from GLOMAP-mode (0.85 – 0.87) agree better with the range
from the aircraft campaigns (0.79 – 0.88) than CLASSIC (0.91). The ambient SSA values
from GLOMAP-mode during the dry season (July – Oct) (0.87 – 0.94) also agree better with
AERONET observations from Alta Floresta and Mongu (Fig. 12a and c). The ambient SSA





also shows a high degree of spatial variability in both models (Fig. 11c and d). These variations are mainly caused by variability of composition and water content. As shown in section 3.5 the hygroscopic growth may be overestimated in both models so the spatial variation of ambient SSA and its relation to humidity may not be entirely realistic. However, the AERONET observations do show a contrast between the drier region of Southern Africa (represented by the Mongu site in Fig. 11c) where the long-term monthly mean SSA drops to 0.82 – 0.85 during July – September, and the moister Amazonian region (represented by the Alta Floresta site in Fig. 11a) where the long-term monthly SSA is around 0.92 during August – September. This observed variation may be explained more by variations in BC content rather than due to variations in hygroscopic growth. There is likely a higher BC content in the aerosol column over Mongu due to the drier vegetation burning more through flaming combustion (some evidence for the higher BC content is found in Fig. 7d for the Welgegund observations that are in the same continental region).

### 3.4.3  Ångström exponent (Å)

The CLASSIC aerosol scheme gives a fairly high Ångström exponent with a dry value of 2.3 for the BBA species (Table 2), and moist values of 1.9 – 2.1 for the fine-mode aerosol mixture over BB dominated regions (Fig. 11e). This is due to the fairly narrow size distribution assumed in CLASSIC. These values of Å are somewhat outside the observed range from the aircraft campaigns (dry values of 1.7 – 2.1 from nephelometer measurements) and AERONET (long-term monthly mean moist values of 1.7 – 1.9). GLOMAP-mode gives slightly lower values of Å than CLASSIC, with dry values ranging from 2.0 – 2.1 (Table 2), and ambient (moist) values ranging from 1.5 – 1.9 over the BB regions (Fig. 11f). These agree quite well with the aircraft observations and AERONET observations during the peak of the burning season (Aug-Sept) (Fig. 12b and d). The seasonal variation of Å observed by AERONET (i.e. the drop to lower values outside the burning season in Fig. 12b & d) is not well captured in either model. This could be due to insufficient representation of coarse particles, such as mineral dust or primary organic particles outside the BB season.

### 3.5  Hygroscopic growth

The hygroscopic growth of aerosol (i.e. the growth of the aerosol with relative humidity due to the uptake of water) leads to enhanced scattering. This can be expressed via the Scattering Growth Factor ($GF_{sca}$), which is the observed or modelled scattering of the aerosol at ambient





humidity divided by the scattering of the same aerosol when completely dried (i.e. at very low
relative humidity). For CLASSIC the hygroscopic growth is specified via an empirical fit that
reproduces the $GF_{sca}$ curve observed by Magi and Hobbs (2003), hereafter MH03. In MH03
$GF_{sca}$ curves were derived from a humidified nephelometer system operated on flights over
Southern Africa during SAFARI-2000. MH03 parameterized the $GF_{sca}$ curves for a range of
aerosol conditions and the CLASSIC scheme uses their "heavy smoke" curve for the fresh
BBA species, and their "regional air" curve for the aged BBA species. These $GF_{sca}$ curves are
shown on Fig 13, along with a representative curve for CLASSIC assuming a mixture with 10
% fresh BBA and 90 % aged BBA. These give a very strong increase of scattering with RH
for the CLASSIC BBA, with $GF_{sca}$ rising to 2.05 at 80 % and to 3.4 at 100 %. With similar
instrumentation Kotchenruther and Hobbs (1998), hereafter KH98, found much lower $GF_{sca}$
for BB dominated aerosol over Brazil (Fig. 13). For RH > 65 % the range from KH98 does
not overlap that from MH03, and at 80 % the range from KH98 is only 1.05 – 1.29. The large
difference between these two observation sets is difficult to reconcile, especially as both were
derived from an airborne humidified nephelometer system. Possibly the regional aerosol
mixture (categorised as "regional air" in MH03) contained a substantial proportion of highly
hygroscopic sulphate from industrial sources in Southern Africa and is therefore not
representative of purely carbonaceous aerosol.
Additional constraints on hygroscopic growth have been provided more recently from
Hygroscopic Tandem Differential Mobility Analyzer (H-TDMA) instruments. A wide range
of measurements, including Amazonian aerosol are summarized in the review of Swietlicki et
al. (2008). More recent measurements for Amazonia are also provided in Whitehead et al.
(2014). In these analyses the hygroscopic growth is summarized via the "kappa" parameter
($\kappa$) that can be used to reconstruct the growth curve from Kohler theory. Swietlicki et al.
(2008) give a range of $\kappa$ values of 0.05 – 0.15 for Amazonian dry season / BB conditions,
leading to $GF_{sca}$ of 1.16 – 1.49 at 80 %. The Kohler curves based on this range of $\kappa$ are also
plotted in Fig. 13. For RH < 90 % the Kohler curves provide an intermediate range of growth
factors that overlap the upper range from KH98 and the lower range from MH03. However,
the Kohler curves have greater curvature and rise very steeply for RH > 80 % and exceed the
range from MH03 for RH > 95 %. This reflects the increasing level of uncertainty in $GF_{sca}$ at
higher RH where growth factors become increasingly difficult to verify from the
observations. Both the empirical fits in KH98 and MH03, and the theoretical Kohler curves
are essentially extrapolated from the observed growth up to 80 % or 90 %.





For GLOMAP-mode the hygroscopic growth curve is calculated based on the Zdanovski-
Stokes–Robinson (ZSR; Stokes and Robinson, 1966) mixing rule. For this comparison we
take the average fine-mode composition from the four regions / sites in Fig 7, which gives a
mixture with 82.6 % organic carbon, 9.4 % sulphate, and 8 % black carbon. The black carbon
is assumed to be hydrophobic whereas organic carbon is assumed hydrophobic when in the
Aitken insoluble mode (where approximately one third of the OA resides) and hygroscopic in
the soluble modes (most of the remaining two-thirds of OA). The water uptake by soluble OA
is based on sulphuric acid but scaled down such that the carbonaceous aerosol from BB takes
up approximately 25 % of the water of an equivalent dry mass of $H_2SO_4$. The $GF_{sca}$ curve in
GLOMAP-mode is capped at a RH of 90 % to avoid overestimation of aerosol scattering and
AODs close to saturation. For relative humidity above 60 % GLOMAP-mode gives lower
$GF_{sca}$ than CLASSIC, with $GF_{sca}$ reaching 1.66 at 80 % and 2.19 for 90-100 % (compared to
2.1 and 2.6 – 3.4 for CLASSIC).  For RH < 60 % GLOMAP-mode has a slightly higher $GF_{sca}$
than CLASSIC and has an unrealistic shape, but this is unlikely to be important compared to
the difference at higher RH.
Overall, although there is large uncertainty from the observations it seems likely that the
CLASSIC scheme overestimates the $GF_{sca}$ and therefore aerosol scattering, AOD and single
scattering albedo for BBA in moist conditions (e.g. RH > 60 %). GLOMAP-mode may also
overestimate the hygroscopic growth, though to a lesser extent. The representation of
hygroscopic growth could be improved in both aerosol schemes. One option would be to use
Kohler curves with observationally constrained $\kappa$ values, though care would be needed in
dealing with the growth assumed at the upper RH range.
**3.6   Vertical distribution of aerosol**
The vertical distribution of BBA in the models depends on the vertical profile of emissions
and on transport and removal processes. The emission profiles and transport processes are
treated identically for the two aerosol schemes but the representation of wet and dry removal
processes are different. The modelled profiles of fine-mode aerosol mass are assessed in Fig.
14 by comparing them with campaign-mean aircraft observations. For the SAMBBA and
DABEX cases the observed profile of fine-mode mass has been estimated from the
nephelometer measurement of dry aerosol scattering multiplied by the fine-mode specific
extinction ($k_{ext,fm}$) and SSA. Due to use of a modified Rosemount inlet serving the
nephelometer on the FAAM aircraft, coarse-mode particles are not well sampled. We



therefore make the assumption that the total nephelometer scattering serves as a reasonable
guide to fine-mode aerosol concentration. For the conversion of scattering to fine-mode mass
we take the $k_{ext,fm}$ and SSA values derived from the in-situ aircraft observations in Table 2.
For SAMBBA (Fig. 14a) the campaign mean profile is representative of the Western
Amazonia region around Porto Velho, Rondonia. The aerosol extinction coefficient derived
from the airborne lidar in SAMBBA was also averaged over a range of flights observing
regional BBA layers in the Amazonian region (Marenco et al., 2015). The lidar-derived
extinction at 355nm was converted to dry extinction at 550nm using an Ångström exponent of
1.7 based on the AERONET September monthly mean at Alta Floresta (Fig. 12a), and the
average humidity growth factor from KH98 (Fig. 13). For DABEX the campaign mean
profile is taken from Johnson et al. (2008a) and included a correction to subtract the scattering
associated with mineral dust aerosol. For SAFARI-2000 no campaign mean profile was
available but Haywood et al. (2003) provides information on the observed range of heights for
the elevated layers observed over the South East Atlantic. To indicate the degree of sampling
error in the mean profiles the standard error is also shown in Fig. 14 for both the observations
and models. For the observations the standard error has been calculated as the standard
deviation of aerosol mass at a given altitude divided by the square root of the number of
profiles (for the nephelometer) or flight sections (for the lidar). For the models the standard
error is calculated as the standard deviation from the ten monthly mean profiles in each
simulations, divided by the square root of ten (the number of years).
The two models predict very similar vertical distributions of fine-mode aerosol with
approximately the same profile shape and magnitude of aerosol mass. In most places
differences between the models are comparable to the standard error associated with
interannual variability. The models also agree quite well with the observations in terms of
reproducing the basic vertical structure and profile shape. Over Amazonia the observed
profile shows a fairly well mixed layer up to 1.5km, a small increase around 1.5- 2 km and
then a gradual decline from 2 – 6 km and very little above 6km. The lidar gives a similar
shaped profile to the nephelometer except with a more pronounced peak around 2km.
Although the concentrations of aerosol mass observed during SAMBBA were highly variable
in space and time (Marenco et al., 2016), the relatively low standard error shows that by
averaging over a sufficient sample of flight sections (lidar) or profiles (nephelometer) the
campaign mean nephelometer and lidar profiles do provide a useful guide for evaluating the
models. The lidar and nephelometer profiles are not expected to match exactly as the spatial



and temporal sampling frequency was different and lidar profiles are more uncertain near the
ground. Both models capture the shape of the observed profiles reasonably well even showing
the increase around 2km. During DABEX the BB dominated aerosol layers were observed to
reside in an elevated layer from 1.5 – 5 km with only low concentrations below. These
elevated layers originated from BB emissions further south but had been undercut by Saharan
air, lofted and transported north and west towards the observed region (mainly around
Niamey, Niger). The models capture the elevated layer but predicted concentrations are lower
than observed. During SAFARI-2000 the BB dominated aerosol over the South Eastern
Atlantic were observed to reside in elevated layers with a fairly consistent layer base at 1.5 +/-
0.6 km and layer top at 4.9 +/- 0.7km. The models both simulate an elevated layer peaking
within this altitude range but with some spread above and below the observed limits of the
layers. The two models give very similar vertical profiles though the mass concentration
peaks at slightly higher values in CLASSIC in the centre of the layer.
Overall the results show that HadGEM3 predicts the vertical profile of BBA quite well
despite the current rather crude set of assumptions for plume injection height. As detailed in
section 2.3 the emissions from Savannah were injected at the surface and emissions from
forest/deforestation uniformly over the lowest 3km. More sophisticated approaches where
plume injection heights are predicted online in the model should certainly be investigated, but
it is encouraging that the current approach works reasonably well for the cases investigated
here.
**4    Conclusions**
We conclude that the implementation of GLOMAP-mode has improved the representation of
biomass burning aerosol in HadGEM3. The modal scheme is able to predict the full aerosol
size distribution, and simulate the variation of aerosol composition and optical properties
giving the scheme increased accuracy over the CLASSIC bulk scheme of HadGEM2-ES. The
simulated aerosol properties, AOD and aerosol vertical distribution are shown to compare
well with observations from SAMBBA and two other aircraft campaigns (DABEX, SAFARI-
2000), and with remote sensing retrievals from MODIS and AERONET.
The analysis of field observations showed biomass burning aerosols to have reasonably
consistent size distributions, Ångström exponents (1.7 – 2.1) and dry specific extinction
coefficients (3.6 – 5.8 m$^2$/g) across different tropical biomass burning regions. CLASSIC
represents this reasonably well by specifying a globally-representative size distribution that



includes the particle size range most important for interaction with solar radiation. GLOMAP-
mode simulated the full size distribution from nucleation to coarse (0.01-10μm), showing
realistic features with good agreement against the available observations. The agreement
between modelled and observed size distributions stems from a well constrained initial size
distribution for the emitted particles, followed by a good representation of how this size
distribution evolves with chemical and microphysical processes. GLOMAP-mode was also
able to predict the optical properties with improved accuracy.
However, the analyses suggest that both aerosol schemes overestimate the uptake of water at
high relative humidity. This overestimation is greater in CLASSIC and is likely to cause an
overestimation of aerosol scattering, AOD and SSA in moist regions. In CLASSIC the aerosol
scattering coefficient rises by a factor of 2.1 from dry conditions to 80 % relative humidity,
whereas in GLOMAP-mode it rises by a factor of 1.7. Although there is considerable
uncertainty and variability amongst observations, recent measurement from   H-TDMA
suggest lower growth factors for aged BB aerosol with the factor of increase in aerosol
scattering in the region of 1.2 – 1.5 from dry to 80 % relative humidity.
The analysis of observations in this study also highlights the strong variations in black carbon
(BC) mass fraction (5 – 12 %) and Single Scattering Albedo (SSA) (0.79 – 0.88) in the
average biomass burning aerosol composition from different tropical source regions. These
variations are a challenge for the models to capture. Whilst the dry BC mass fraction and SSA
in GLOMAP-mode (7-10 %; 0.85 – 0.87) are closer to the observed values than CLASSIC (5-
9 %; 0.91), the variability between source regions is lower than observed. This may point to
the need for a wider range of BC:OC ratios in the emissions data, which in GFED3 are based
on Andreae and Merlet (2001). These have been updated in GFED4 (Giglio et al., 2013)
based on Akagi et al. (2011) and future studies may provide useful feedback on whether these
improve the variability of aerosol composition in models. The emissions of BBA had the
same prescribed vertical profile in both models and led to very similar vertical distributions of
fine-mode aerosol mass over the main tropical BB regions that compared well with the
airborne in-situ and lidar observations.
Whilst both schemes gave good agreement between observed and modelled AODs over BB
regions, this was achieved by scaling up the total aerosol emissions from GFED3.1 by a
global scaling factor of 1.6 for CLASSIC and 2.0 for GLOMAP-mode. This might suggest
that the emissions of BC and OC from GFED3 lead to an underestimate of the aerosol mass.





However, we note that there is considerable uncertainty in other parameters in the models that
affect the aerosol mass and AOD from BB sources. Firstly, there is considerable uncertainty
in the ratio used to convert the OC (i.e. carbon mass provided by the emissions data) to the
total mass of POM emitted in the models. This depends on the ratio of carbon to oxygen and
other elements in the emitted aerosol. In the current configuration of HadGEM3 CLASSIC
does not account for this issue (effectively neglecting the non-carbon mass) and GLOMAP-
mode converts the OC to POM using a ratio of 1.4 that is likely too low for biomass burning
emissions. On the other hand CLASSIC increases the total aerosol mass by a factor of 1.62 on
a 6 hour e-folding timescale to represent condensation growth during ageing (a process that
GLOMAP-mode does not include). Therefore, the emission scaling factors required to
generate agreement between modelled and observed AODs clearly depend on these other
scaling applied within the aerosol schemes, as well as aerosol optical properties and aerosol
lifetime. Other models may not require emission scaling to gain good agreement with
observed AODs or may require different scaling factors outside the range 1.6 – 2.0 found in
this study. It is also worth noting that there are large differences between emission factors
estimated for difference measures of the aerosol mass: BC + OC, Total Carbon (TC), Total
Particulate Matter (TPM), PM2.5 and PM10 (see Andreae and Merlet 2001; Akagi et al.,
2011). For instance the emission factors for TPM are a factor of 2.3 – 2.4 higher than the sum
of BC + OC in GFED3 (based on Andreae and Merlet 2001) for tropical BB sectors. Using
TPM instead of BC and OC in our simulations would therefore have led to an overestimation
of AOD in tropical regions unless the global emission scaling factor was reduced to
approximately 0.67 in CLASSIC and approximately 1.15 in GLOMAP-mode. With such large
uncertainty and observed variability in emission factors, POM:OC ratios, hygroscopic growth
and secondary formation of organics, it is difficult to advocate any particular set of changes
that would improve the models, though clearly there is scope to reduce the discrepancy
between modelled and observed AOD without the use global emission scaling factors.
Furthermore, although tuning the emissions gave good overall agreement with observed AOD
in the dominant tropical BB regions, some regional discrepancies remained. In particular, we
note a low bias over West Africa and a high bias over the Congo basin during Northern
hemisphere winter. The AOD over South East Asia and Northern Australia during their BB
seasons were also underestimated in our simulations, but the contribution of BB to AOD in
the high latitude Boreal forests seems to be slightly overestimated. Regional biases in AOD
may be caused, to some extent, by regional (or biome specific) biases in the total emission



rate. Other factors may include variations in aerosol optical properties between different
regions (e.g. due to different size distribution or water uptake) that may not be captured in the
models. Applying a globally uniform scaling factor to account for current uncertainties in BB
emission datasets is therefore not sufficient to reconcile the modelled AOD with observations.
GFED version 4 (Giglio et al., 2013) has already made significant progress in addressing
biases related to small fires (Randerson et al., 2012) that are difficult to identify from burned
area products. Follow on studies from this work are recommended to assess the impact of
recent developments in fire emission modelling on reducing such regional biases.
Overall we conclude that GLOMAP-mode provides a good simulation of BB aerosol for
modelling their impacts on radiation and climate. Impacts on CCN and cloud microphysics
have not been evaluated here but have been assessed previously in Bellouin et al. (2013). This
study does show clear improvements to the aerosol size distribution and composition in
GLOMAP-mode that is important for aerosol indirect effects. This shows the benefits of
including a more detailed representation of aerosol microphysical and chemistry processes.
However, the model could merit from further improvements to BB processes, including more
accurate estimates of the emission flux, the composition of emitted particles (which can vary
considerably with vegetation / fuel type), and the injection height profile. We also note large
uncertainties in the representation of hygroscopic growth, ageing, and absorption (including
the role of brown carbon). This is partly due to the complexity of these processes and
difficulties in constraining them with observations.
**Author contributions**
J. Langridge, E. Darbyshire, W. Morgan, K. Szpeck, J. Brooke, F. Marenco contributed
towards the analysis of FAAM aircraft observations from SAMBBA, J. Haywood, H. Coe, P.
Artaxo, K. Longo were co-principal investigators on the SAMBBA project, J. Mulcahy, G.
Mann, N. Bellouin, M. Dalvi contributed to the implementation of GLOMAP-mode aerosol
scheme in HadGEM3.
**Acknowledgements**
The Facility for Airborne Atmospheric Measurement (FAAM) BAe-146 Atmospheric
Research Aircraft is jointly funded by the Met Office and Natural Environment Research
Council and operated by DirectFlight Ltd. We would like to thank the dedicated efforts of



FAAM, DirectFlight, INPE, University of Sao Paulo, and the Brazilian Ministry of Science
and Technology in making the SAMBBA measurement campaign possible. For AERONET
data we thank the PI investigators and their staff for establishing and maintaining the sites
used in this investigation (Alta Floresta and Mongu: Brent Holben, Ilorin: Rachel T. Pinker,
Chiang Mai: Serm Janjai, Bonanza Creek: John R. Van de Castle, Jabiru: Ross Mitchell). We
thank Andrew Sayer and Robert Levy from Goddard Space Flight Centre for their advice with
MODIS aerosol products. We thank Ville Vakkari for help in selecting data from the
Welgegund station. JMH, ED, WTM, HC, GM and NB were funded by SAMBBA (NERC
grant NE/J009822/1). BJ, JMH and JM were funded under the Joint UK DECC/DEFRA –
Met Office Hadley Centre Climate Programme (GA01101). JMH was part funded by the
IMPALA grant (NE/M017214/1) via Future Climates for Africa (FCA) funding provided by
NERC and DFID.

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



| Aerosol scheme | Emitted particle properties | | | Scaling factors applied | | |
|---|---|---|---|---|---|---|
| | Dg | σ | BC:OC | POM:OC conversion | Ageing growth factor | Global emission scaling |
| **CLASSIC** | 0.20 | 1.3 | 0.093 | n/a | 1.62 | 1.6 |
| **GLOMAP-mode** | 0.15 | 1.59 | Variable (GFED3.1) | 1.4 | n/a | 2.0 |

2   **Table 1.** Biomass burning aerosol emissions: emitted particle properties and scaling factors

3   applied.





| Data source | Campaign / region | SSA | Å | $k_{ext,fm}$ (m²/g) | g | References |
|---|---|---|---|---|---|---|
| *models* | | | | | | |
| **CLASSIC aged BBA** | Global | 0.91 | 2.3 | 5.0 | 0.58 | Haywood et al. (2003) |
| **GLOMAP-mode fine-mode** | SAMBBA West (Rondonia) | 0.87 | 2.0 | 4.8 | 0.63 | |
| | SAMBBA East (Tocantins) | 0.86 | 2.1 | 4.5 | 0.60 | |
| | DABEX (West Africa) | 0.85 | 2.0 | 4.6 | 0.61 | |
| | SAFARI (S. Africa) | 0.86 | 2.0 | 4.8 | 0.62 | |
| *Observations* | | | | | | |
| **In-situ aircraft observations** | SAMBBA West (Phase 1, Rondonia)[b] | 0.88 +/- 0.05 | 1.9 +/- 0.3 | 3.6 +/- 0.06 | [c]0.66 +/- 0.06 [d]0.59 +/- 0.05 | Darbyshire et al. (in preparation, 2016b), Brooke (2014) |
| | SAMBBA East (Tocantins) | 0.79 | 2.1 +/- 0.2 | n/a | 0.57 +/- 0.05 | Darbyshire et al. (in preparation, 2016b). |
| | DABEX (West Africa) | 0.81 +/- 0.05 | 1.7 | 5.8 | 0.63 | Johnson et al. (2008) |
| | SAFARI-2000 (S. Africa) | 0.88 +/- 0.04 | n/a | 4.3 | 0.58 | Haywood et al. (2003)[e] |

**Table 2.** Dry aerosol optical properties at 550nm from model and observations including Single Scattering
Albedo (SSA), Ångström exponent (Å), fine-mode specific extinction coefficient ($k_{ext,fm}$), and asymmetry
parameter (g). Error bounds are given to observed parameters, where available, to reflect uncertainty in
the measurement.





[a] Assuming a representative mixture with 10 % fresh and 90 % aged BB aerosol.
[b] Phase 1 of SAMBBA was from 14 – 22 September 2012.
[c] Calculated from the nephelometer backscatter fraction based on Andrew et al. (2006).
[d] Derived from Mie calculations in Brooke (2014)
e Haywood et al. (2003) results for SSA reassessed in Johnson et al. (2008).

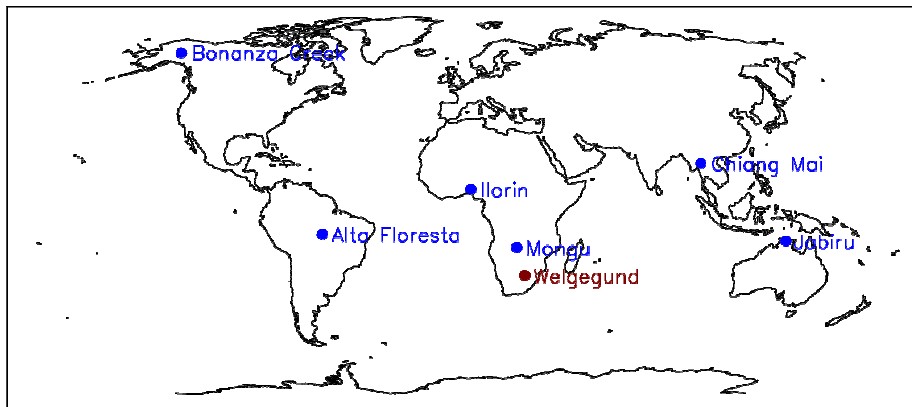

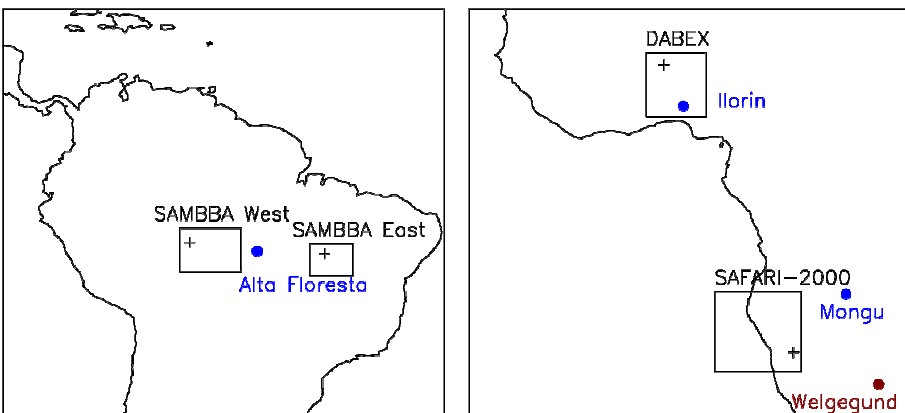

**Figure 1.** Maps showing the location of AERONET sites (blue), the Welgegund ground
station, and the averaging boxes used corresponding to the flight regions from SAMBBA
(West and East), DABEX and SAFARI-2000. Plus symbols indicate the locations of the main
airbases used for the flights: Porto Velho for SAMBBA West, Palmas for SAMBBA East,
Niamey for DABEX, and Windhoek for SAFARI-2000.





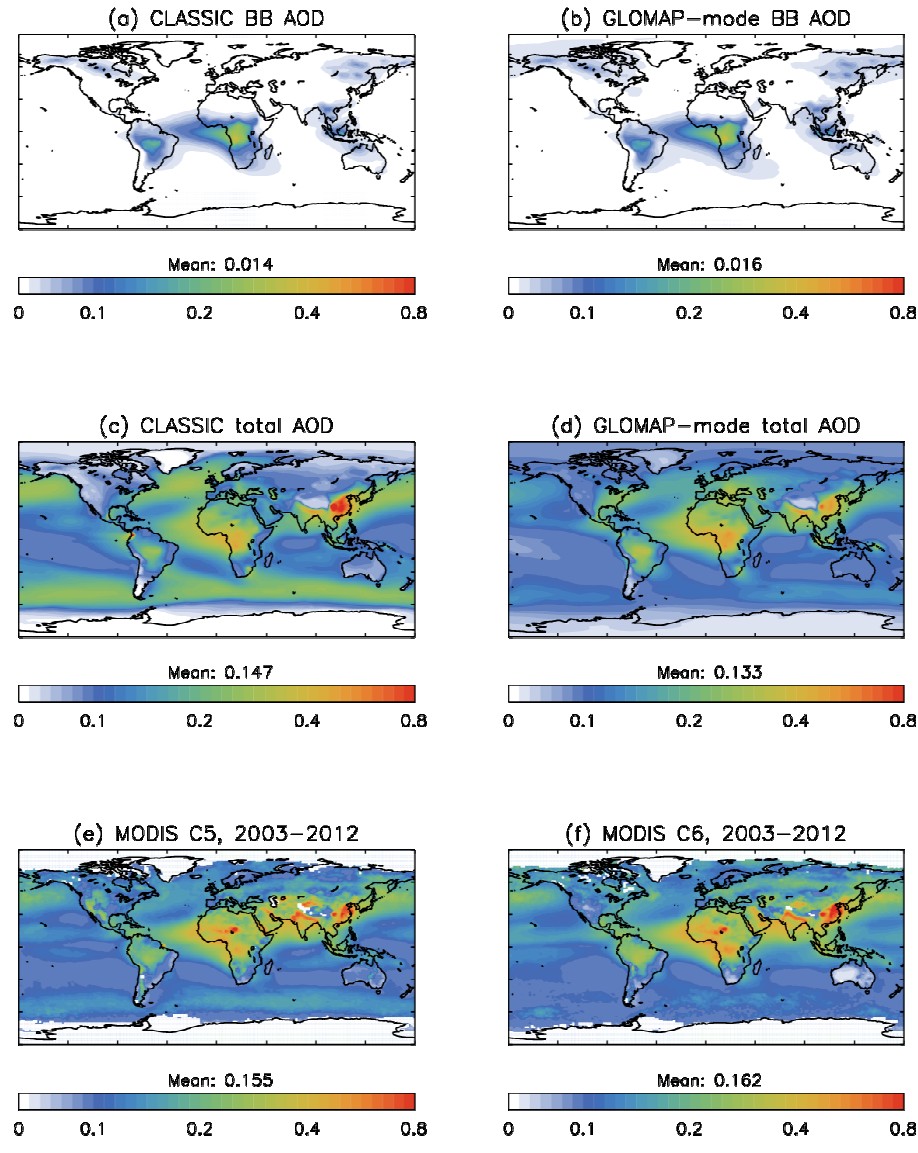

2  **Figure 2.** Decadal mean AOD at 550nm from the model simulations and MODIS. BB AOD

3  is the contribution of BBA emissions to the total AOD. Model means from 2002 – 2011 and

4  MODIS from 2003 – 2012 based on Aqua.





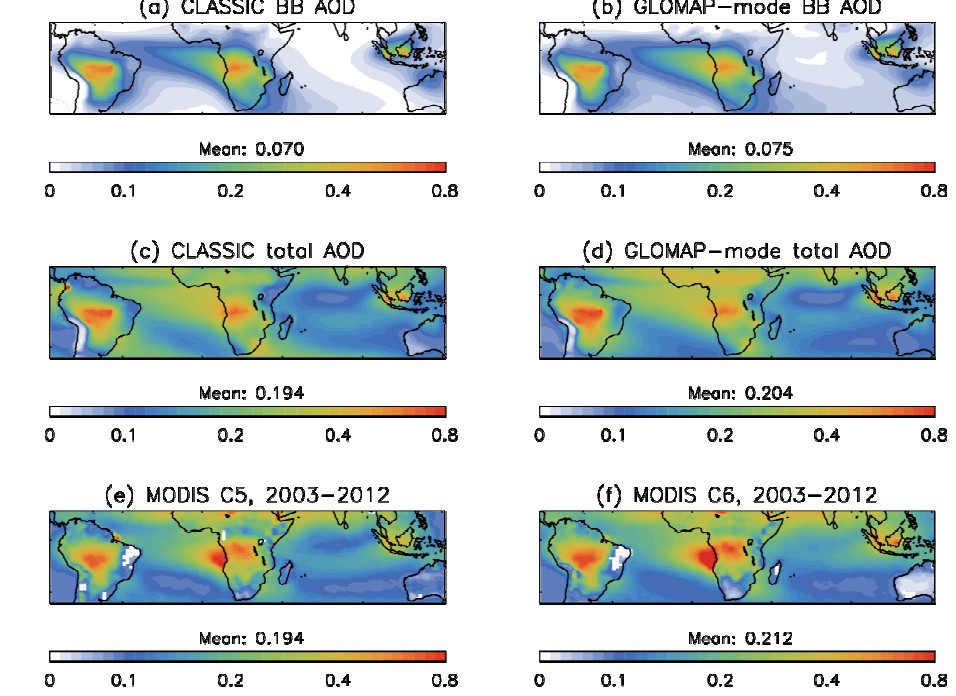

2    **Figure 3.** Same as Fig 2 but for the month of September.





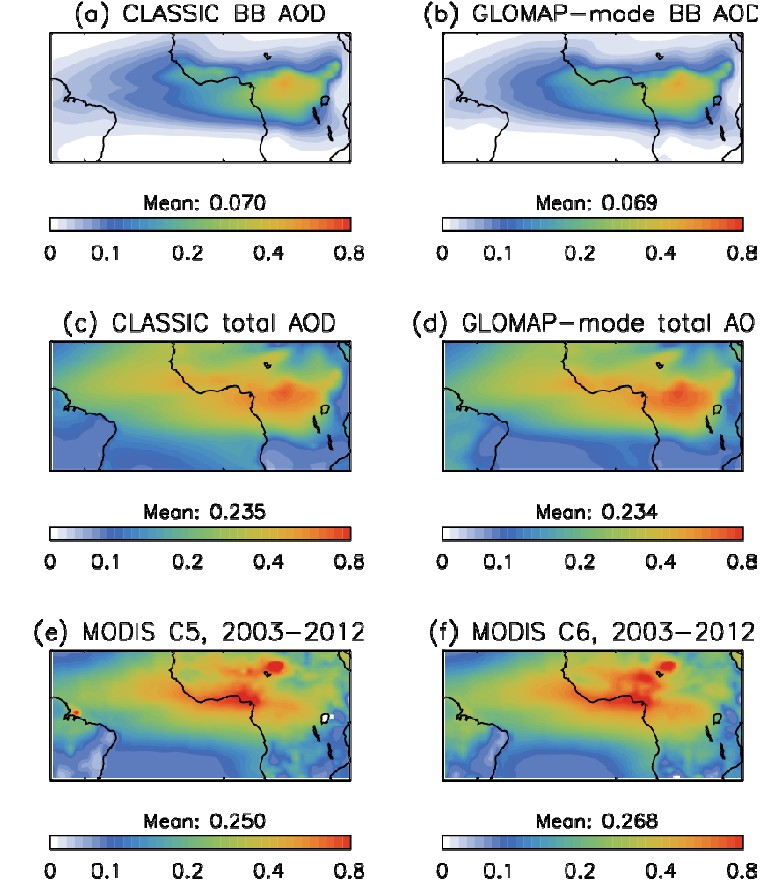

**Figure 4**. Same as Fig 2 but for the month of January.



**Figure 5.** Monthly mean AOD at 550nm from six AERONET sites (grey squares), and the

same locations from GLOMAP-mode (blue) and CLASSIC (red). The contribution to AOD

from BBA is shown by dashed lines.







**Figure 6.** Modelled fine-mode aerosol composition from HadGEM3 for CLASSIC and
GLOMAP-mode (including sulphate, BC and OA only). Plots show: (a, b) fine-mode
mass burden (mg m$^{-2}$), (c, d) mass fraction of OA (%), (e, f) mass fraction of BC (%), (g,
h) mass fraction of sulphate (%). Stipples indicate grid columns where more than 75 % of
the fine-mode aerosol mass originates from biomass burning emissions (based on the
speciation in the CLASSIC simulation). Mean values beneath each plot give the average from
grid columns marked by these stipples.





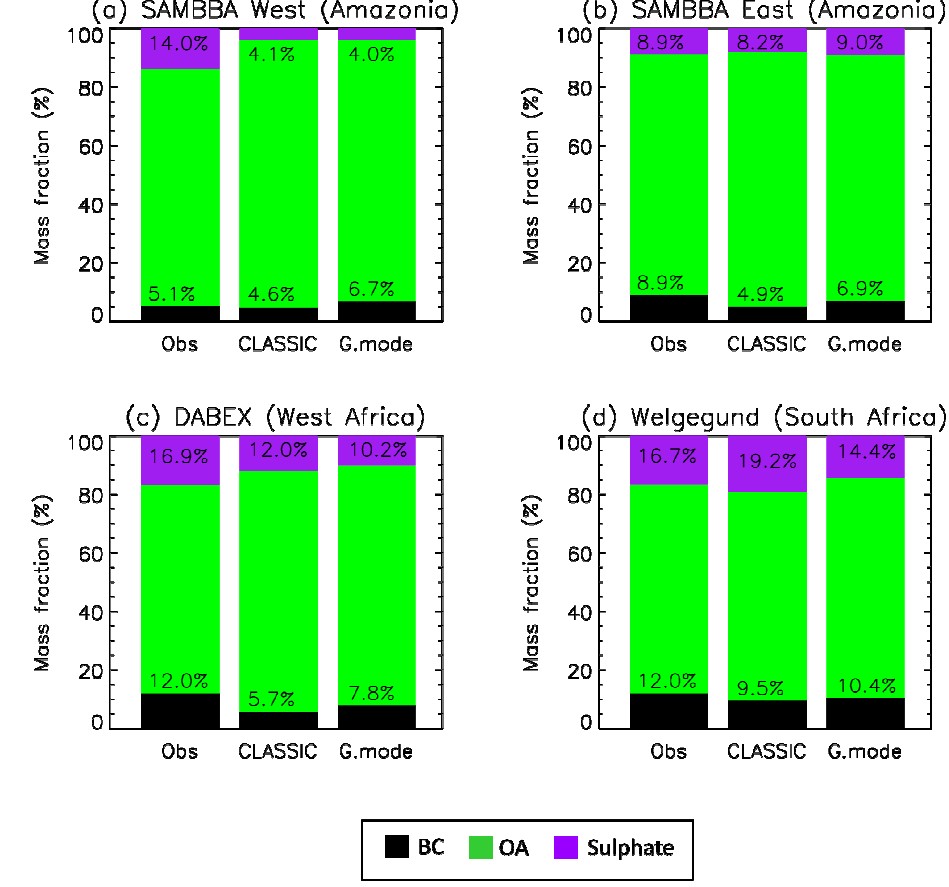

**Figure 7.** Mass fractions (%) of black carbon (black), organic aerosol (green), and
sulphate (purple) (excluding other fine-mode aerosol components). Observed data are
monthly averages from field campaigns including: SAMBBA (Amazonia, September
2012), DABEX (West Africa, Jan 2006), and the Welgegund site (South Africa,
September 2010). Modelled data are long-term monthly mean values corresponding to
the month and location of the observations. Welgegund model data is for aerosol
composition at the surface (lowest model level), SAMBBA and DABEX model data is
averaged over 0 – 5km. The BC and sulphate mass fractions are labelled on each bar.





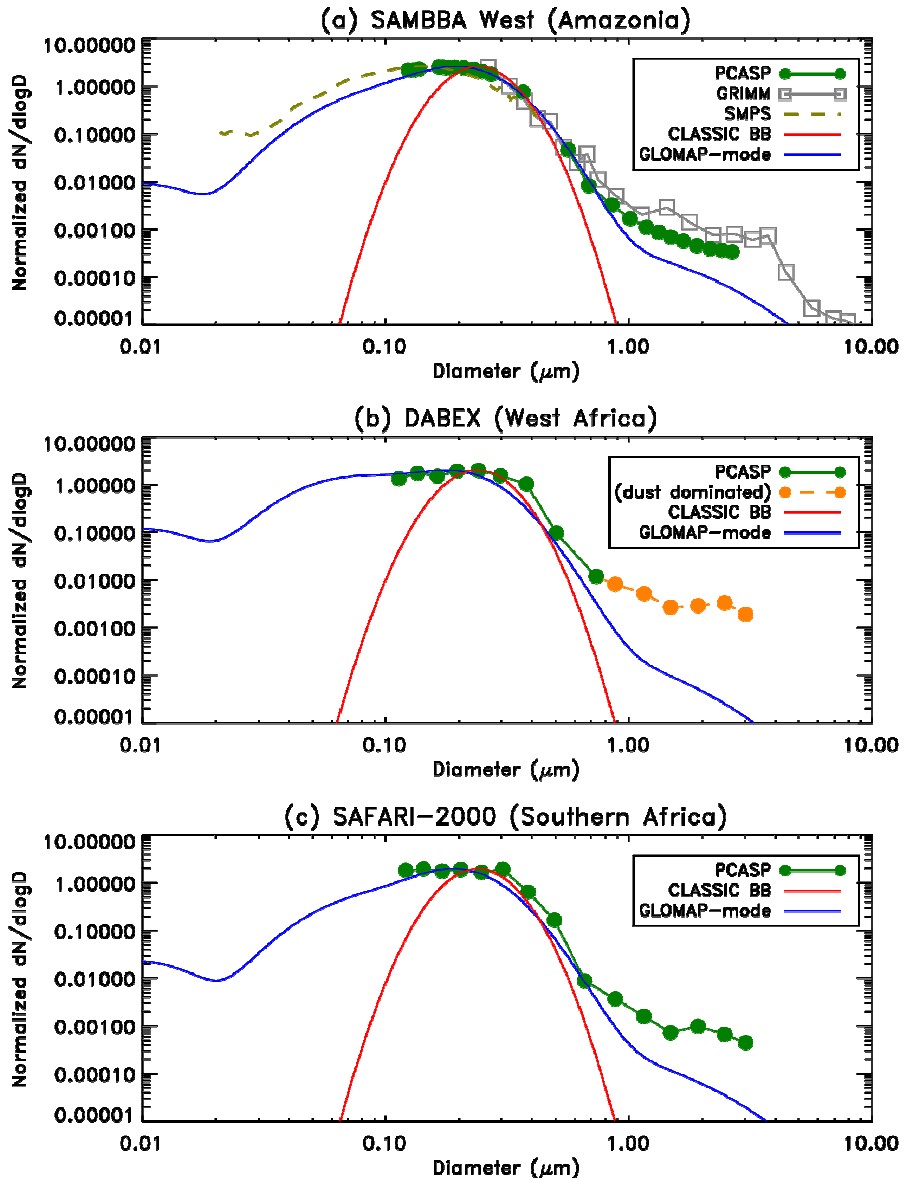

**Figure 8.** Aerosol number size distributions (dN/dlogD) versus particle diameter from aircraft

observations (PCASP, GRIMM, SMPS) showing the mean distribution from three campaigns.

CLASSIC curve is a representative mixture of 10 % fresh and 90 % aged BBA species,

GLOMAP-mode is the complete size distribution over all 5 modes averaged over the flight

regions in Fig. 1 and over 0 – 5km.





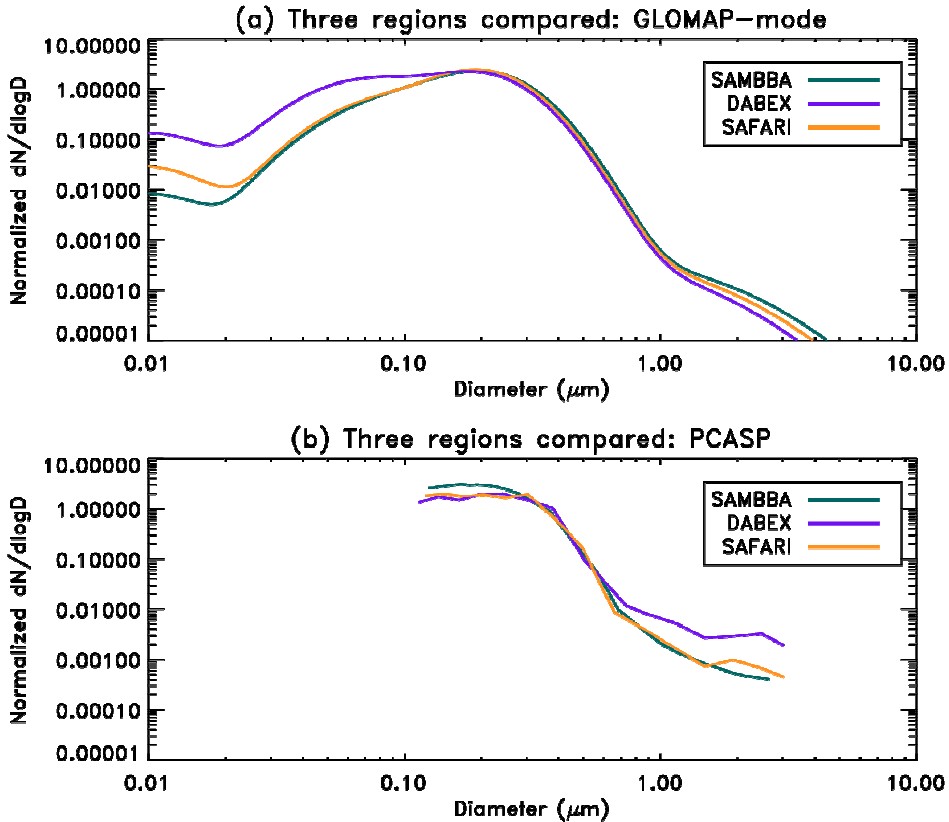

2    **Figure 9.** Same as Fig. 7 but showing only GLOMAP-mode curves and PCACP data.





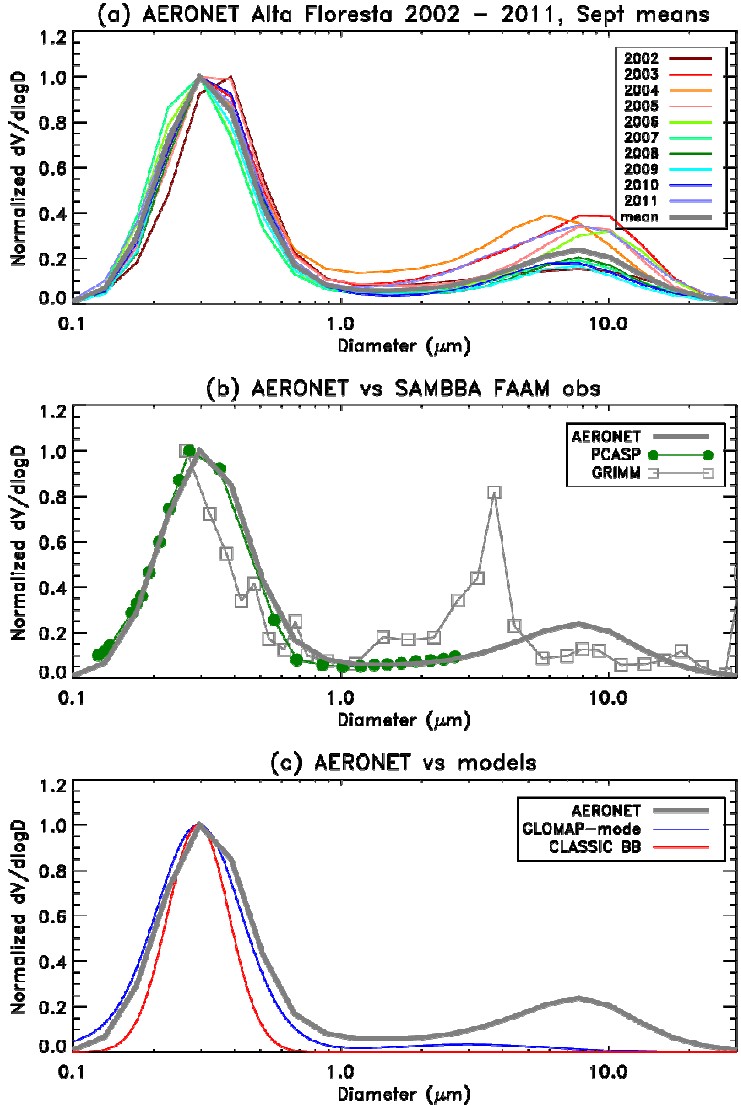

**Figure 10.** Aerosol volume size distributions (dV/dlogD µm$^3$/µm$^2$) vs. particle diameter for
(a) September means from AERONET Alta Floresta (Southern Amazonia) for 2002-2011
along with the long-term monthly mean from all years, (b) Comparison of AERONET 10-
year September mean with FAAM averages from SAMBBA West region, normalized by peak
concentration, (c) Comparison of AERONET 10-year September mean with HadGEM3
September monthly mean output, column-integrated mean over Alta Floresta for GLOMAP-
mode (all active size modes) and CLASSIC (BB species only).





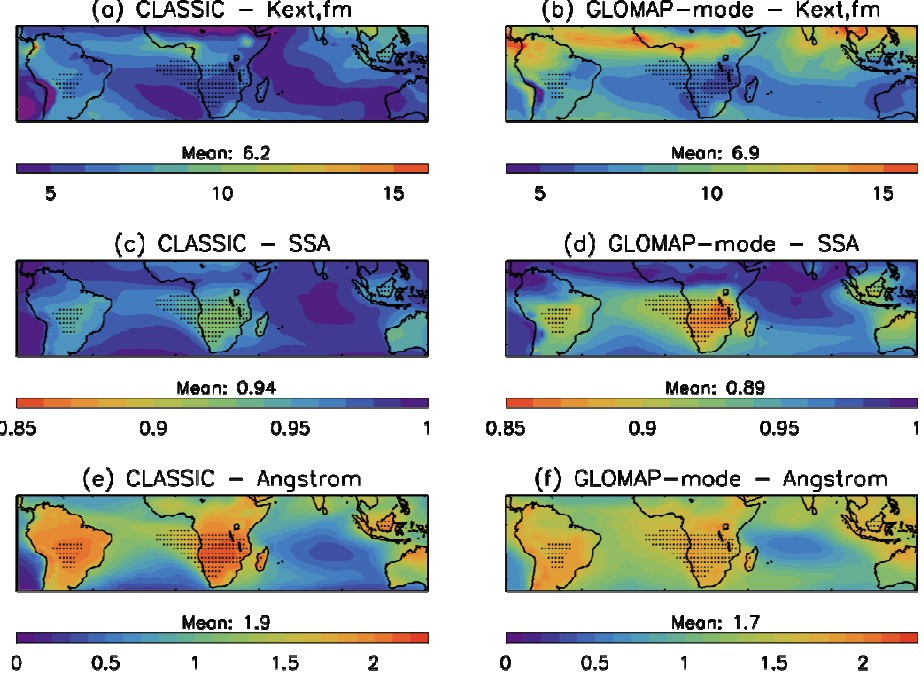

**Figure 11.** Column average moist aerosol optical properties from CLASSIC and GLOMAP-mode for September long-term monthly mean. Properties are the fine-mode specific extinction coefficient ($k_{ext,fm}$), Single Scattering Albedo (SSA) and Ångström exponent. Stipples indicate grid columns where more than 75 % of the fine-mode aerosol mass originates from biomass burning emissions (based on the speciation in the CLASSIC simulation). Mean values beneath each plot give the average from grid columns marked by these stipples.



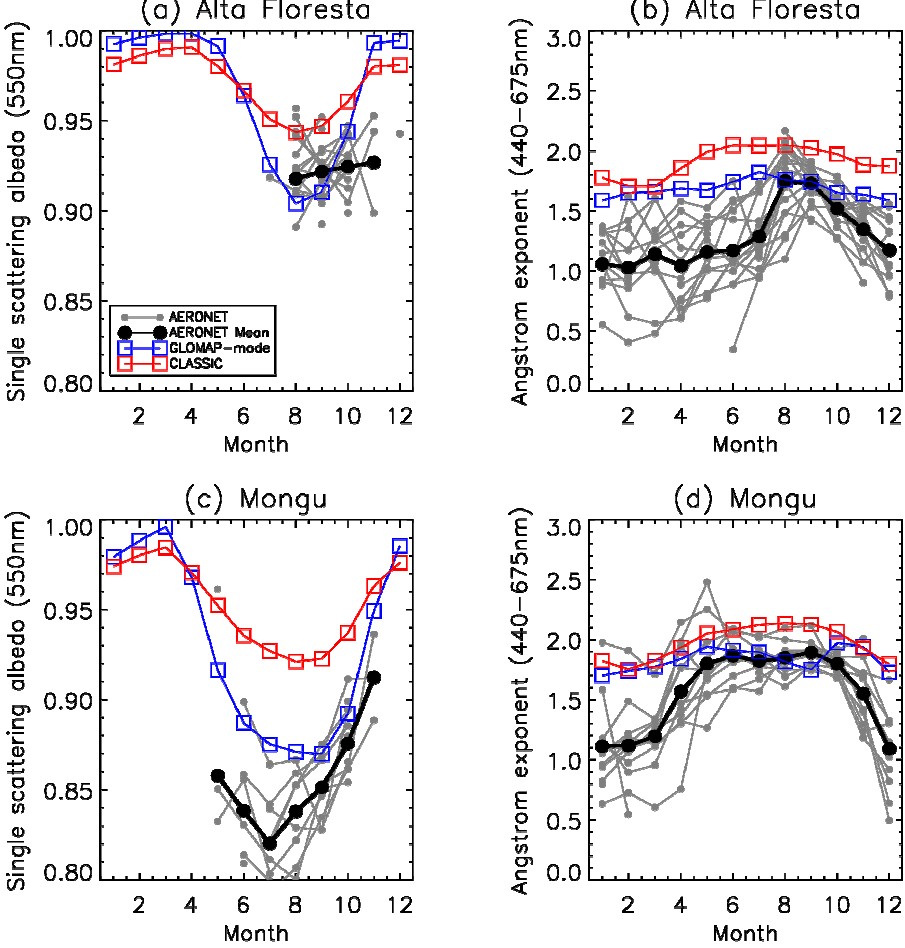

**Figure 12.** Seasonal cycle of moist aerosol optical properties (single scattering albedo and Ångström exponent). AERONET data from Alta Floresta (Southern Amazonia) and Mongu (Southern Africa) include all available monthly means (grey) and the long-term monthly mean (black) for months with good data coverage (see text). Co-located model data from GLOMAP-mode (red) and CLASSIC (blue) are shown taking the column average long-term monthly means.





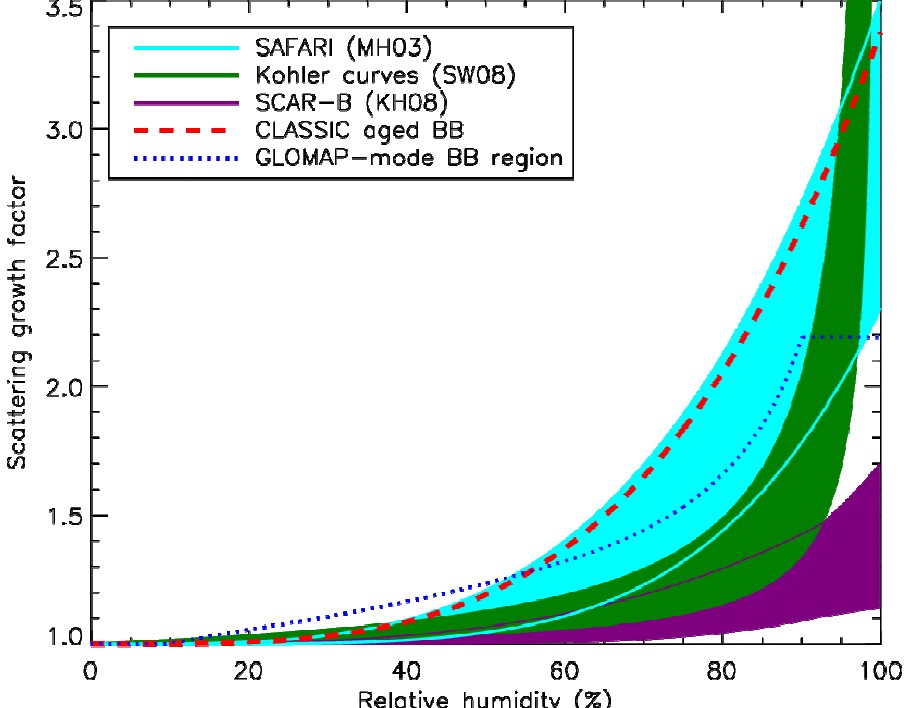

**Figure 13.** Hygroscopic growth curves showing the increase in aerosol scattering at 550nm

with ambient relative humidity from a variety of observational sources and from the models.

The curve for CLASSIC assumes a mixture of 10 % fresh and 90 % aged BBA. The curve for

GLOMAP-mode is calculated based on the average composition from the four BB regions in

Fig. 7. The solid filled areas show the range of growth factors estimated from each

observation source (see text).





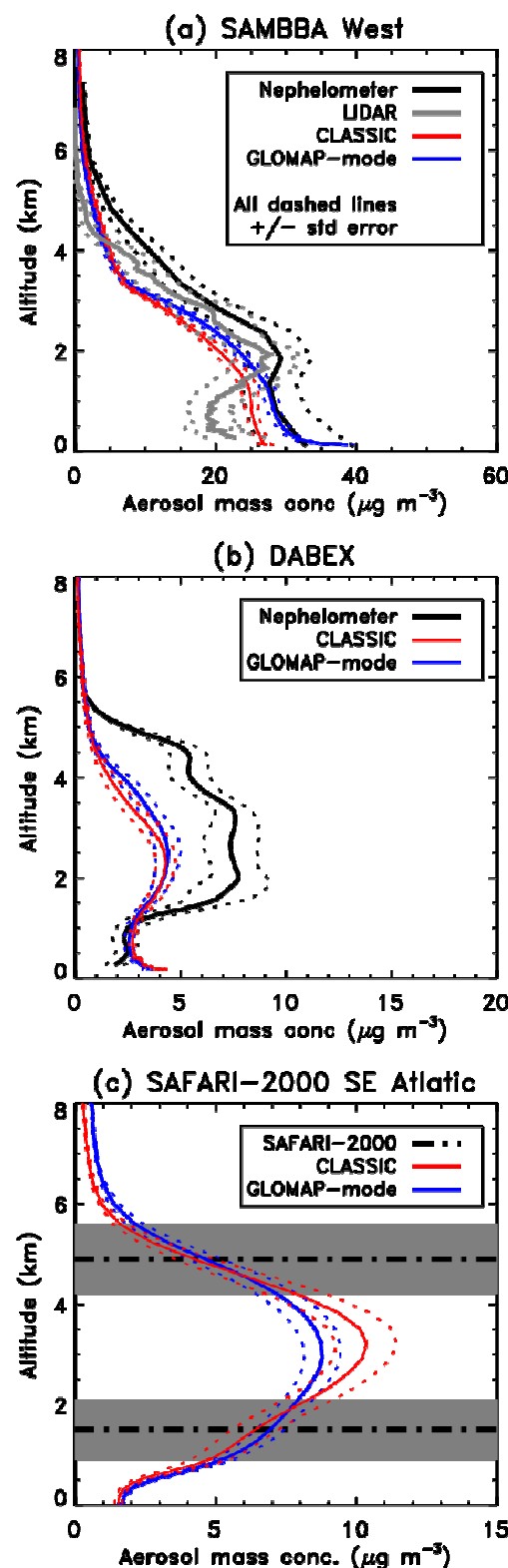





**Figure 14.** Vertical profiles of fine-mode aerosol mass concentration for the SAMBBA,
DABEX and SAFARI-2000 airborne campaigns, including model averages for CLASSIC
(red) and GLOMAP-mode (blue). Dashed lines show the mean +/- the standard error. Profiles
of mass concentrations have been estimated from campaign-averaged nephelometer (black)
and lidar (green) observations using the fine-mode specific scattering ($k_{sca,fm}$) and extinction
coefficients ($k_{ext,fm}$), respectively, derived from the in-situ aircraft observations (see the
second half of Table 2; $k_{sca,fm} = k_{ext,fm} * SSA$). The SAFARI-2000 observations indicate the
average altitude of BBA layer base and top (black dot-dashed line) +/- the standard deviation
(grey shading).