# Peer review of "Evaluation of biomass burning aerosols in the HadGEM3"

_Atmospheric Chemistry and Physics, 2016_

## Referee Comment (RC1) · Anonymous Referee #1 · 11 Jul 2016

This well-written, scientifically sound paper uses observations from the South American Biomass Burning Analysis (SAMBBA) campaign to evaluate biomass burning aerosol properties in then HadGEM2 climate model. Two aerosol modules are considered: a simpler mass-based model (CLASSIC) and a more complex microphysical aerosol module (GLOMAP). The authors show that both schemes reasonably represent biomass burning aerosol properties once emissions are scaled up, and that scaling depends on the scheme considered. Overall, the microphysical representation of the aerosol produces a slightly superior biomass burning aerosol representation, particularly in the representation of optical property variability and size distribution.

I find this study acceptable for publication in ACP after the authors address a few minor

points, which I outline below.

1. Page 2, Line 19: Space between needed: "3km" -> "3 km" 2. Page 3, Line 3 (and thereafter): "Biomass Burning" -> "biomass burning" 3. Page 3, Line 5: Remove quotations around aerosol-radiation interactions. 4. Page 3, Line 7: I do not agree with the statement that aerosol absorption suppress the hydrological society. What about the Elevated Heat Pump effect of Lau et al., Randles et al. etc. and similar studies? In some cases absorption suppresses precipitation, in others it can enhance precipitation. Please reword to state it can have either impact, depending on environmental conditions. 5. Page 6, Line 14: This is unclear. Are then same exact oxidant fields (climatology) used for both the CLASSIC and GLOMAP runs? If not, why? Wouldn't this make a difference? 6. Page 7, Line 15: This seems odd. No sea salt transport over land? You just wash it out once it crosses the coastline? Is this realistic? 7. Page 7, Line 7: I'm a bit confused about how emissions are prescribed in CLASSIC. Doesn't the OC:BC ratio on emission come from the emissions inventory (GFED)? Please clarify. 8. Page 7, Line 25 (and thereafter): Put a space between number and unit (micrometer). 9. Page 7, Line 31: "Refractive Index" -> "refractive index" 10: Page 9, Line 14: "hygroscopic" -> "hydrophilic" 11. Page 9, Line 16: Do you get optical properties on the fly or from a pre-computed look-up table? 12. Page 9, Line 19: You don not consider UV absorption (brown carbon)? Why not? Can you indicate if this should be the case here? Does SAMBBA lend support for a representation of brown carbon? 13. Page 9, Line 29: You imply that QFED and FEER use global scaling. They do not. QFED, for example, uses biome-specific scaling. Please reword so that is clear while these inventories do scale their emissions, the scaling is more complex then multiplying by a single number globally as you do here. Considerable effort was made by these inventories in their scaling efforts. 14. Page 10, Line 1: Why did you choose to use GFED as opposed to QFED or FEER? Curious. 15. Page 10, Line 30: When calculating optical properties, do you use the POM or OC mass? 16. Page 14, Line 7: Why do you just consider Aqua? 17. Page 14, Line 16: Are you using the Level 3 product? Is this the best to be using? Do you do any sub-sampling of your model to only consider satellite

observation times? Would sub-sampling impact your comparison? You do say this on page 16 but I would comment here as well. 18. Page 16, Line 28: You say the model AOD is calculated on clear-sky RH. What does this mean? Does your model track 2 RH (clear- and cloudy- sky)? Or do you represent RH as a sub-grid PDF? Please explain. 19. Page 17, Line 28: Ilorin will also be impacted by dust aerosol; is it therefore correct to conclude that your under-representation is due to biomass burning aerosol alone? Or could your dust simulations also be off? 20. Page 17, Line 32: How are you attributing this to BB aerosol? Couldn't it also be anthropogenic? 21: Page 18: Line 15: OA = POM? 22. Page 19, Line 2: I still don't understand why both CLASSIC and GLOMAP do not have the BC:OA ratio coming from the emissions inventory (since that is built into the emissions). 23: Page 20, Line 9: Remove space before % 24: Page 22, Line 24: "Sampling issues (e.g. altitude) may be a large source of representativeness error in the ... " 25: Page 23, Line 25: Do you not also sub-sample the model at AERONET observation times? Why not? Would't that matter? 26: Page 24, Line 27: Comment on why it's expected that a microphysical model can achieve more realistic variability in optical properties than a mass-based model. This is an important point. 27. Page 25, Line 30: "Scattering Growth Factor" -> "scattering growth factor" 28. Page 30, Line 8: This is a very important point. Can you comment on why the growth factor is smaller in the observations? Do any of the chemical data suggest reasons for this? Do you think this is characteristic of all BB or just over south America?

29: Figure 3: Could remove repeated colorbars (since all are the same) and just have one large one. What is the white in the squares in (e)? Missing data? Maybe grey this out. 30: Figure 5: For AERONET, put the standard deviation on the month to indicate sub-monthly variability Could also do this for the model as shading. 31: Figure 8: The part labeled "dust-dominated" -> is there another measurement that confirms this? You see the same thing (almost) in SAFARI-2000 and there was not as much dust in southern Africa correct? 32. Figure 11: Again, one colorer for each 2 panels across. 33. Figure 12: Rather than all the grey spaghetti lines, why not just shade max-min? Would be easier to see

---

## Referee Comment (RC2) · Anonymous Referee #2 · 1 Aug 2016

The authors present an evaluation of the performance of two aerosol simulation modules used within HadGEM, against a range of observations. The manuscript is thorough and well written, and falls within the scope of the journal. Some of the arguments made require some further clarification or quantification. After this and some other minor issues are dealt with, I recommend publication in ACP.

Main comments

My main question is about the scaling factors of GFED emissions used. I understand that they were chosen so as to "match the magnitude of AOD observations" (p16,l7) / "give good agreement between modelled and observed mid-visible AOD" (p9,l27). However, no further description of how the exact scalings were chosen is given, and

yet the scaling factors make it all the way into the abstract, where they read as a conclusion about emission underestimates by GFED. Even slightly different scaling factors might change the discussion of AOD comparisons in section 3.1, in particular regional features such as discussed e.g. on page 16, lines 18-20. From the global/regional AOD values given in Figures 2-4, which reflect the entire aerosol distribution in the model, it seems that there's some range of scaling factors that might fit (as the authors also discuss on page 31. I'd suggest adding some sensitivity studies here, quantifying (through model bias calculations or similar) that any AOD comparison improvement in GLOMAP over CLASSIC is robust, and maybe toning down the focus on the scaling factors to be more in line with the authors own discussions in the Conclusion section.

Further, it's difficult to properly see from Figures 2-4 what regions are most affected by the improved aerosol treatment. I suggest adding some difference or ratio plots (maybe just versus one of the MODIS collections) to highlight the changes.

Figure 7 is another example where I don't quite follow the authors argument that GLOMAP is a clear improvement (page 19-20). It seems that some observed species ratios are closer to CLASSIC, some to GLOMAP, and unifying MAC to 10 m2/g also seems to pull both ways. Please add some quantification of the improvement here.

As the authors themselves point out (p29,l17), sensitivity tests of the injection height assumptions of BB aerosols would be very useful - both for the present analysis, and for the aerosol community. Adding 1-2 simulations here would further increase the relevance of the paper.

Minor/technical issues

- Many references are made to two as-yet unavailable manuscripts ("Darbyshire et al., in preparation"). As comparisons to SAMBBA is a main point of the paper, and topics such as data averaging are important for reproducibility, this was somewhat annoying when trying to understand the analysis presented here.

- P5l21: Please give a brief explanation of the MetUM, even if it's probably known to most readers.

- P6l20: So SO4 and BC emissions are climatologies, but BB emissions have annual variations? Or do they change through interpolations? Please clarify.

- P7l15: Will the assumption of no sea salt transport over land in CLASSIC affect AOD comparisons in BB-emission coastal regions? Does this contribute to differences to GLOMAP?

- P12l1: Add "of" between "prevalence" and "moist".

---

## Author Comment (AC1) · 29 Sep 2016

**Response to comments from reviewer #1:**

**"Evaluation of biomass burning aerosols in the HadGEM3 climate model with observations from the SAMBBA field campaign"**

Reviewer comments in normal font

*Responses in italics,* **change to manuscript text shown in bold**

This well-written, scientifically sound paper uses observations from the South American Biomass Burning Analysis (SAMBBA) campaign to evaluate biomass burnin aerosol properties in then HadGEM2 climate model. Two aerosol modules are considered: a simpler mass-based model (CLASSIC) and a more complex microphysical aerosol module (GLOMAP). The authors show that both schemes reasonably represent biomass burning aerosol properties once emissions are scaled up, and that scaling depends on the scheme considered. Overall, the microphysical representation of the aerosol produces a slightly superior biomass burning aerosol representation, particularly in the representation of optical property variability and size distribution. I find this study acceptable for publication in ACP after the authors address a few minor points, which I outline below.

*Thank you for the positive feedback*

1. Page 2, Line 19: Space between needed: "3km" -> "3 km"

2. Page 3, Line 3 (and thereafter): "Biomass Burning" -> "biomass burning"

3. Page 3, Line 5: Remove quotations around aerosol-radiation interactions.

**Done** *and made consistent through manuscript*

4. Page 3, Line 7: I do not agree with the statement that aerosol absorption suppress the hydrological society. What about the Elevated Heat Pump effect of Lau et al., Randles et al. etc. and similar studies?In some cases absorption suppresses precipitation, in others it can enhance precipitation. Please reword to state it can have either impact, depending on environmental conditions.

*Yes, this is true. Strong absorption can sometimes increase precipitation regionally. The sentence has been expanded to make this point and I add the two citations the reviewer points to.*

***Page 3, Line 7:***

***"These effects may suppress the hydrological cycle by stabilizing the lower troposphere, although strong absorption can in some cases enhance precipitation regionally by increasing low-level convergence (Wu et al., 2013; Ramanathan et al., 2001; Lau et al., (2006); Randles et al., 2008)."***

5. Page 6, Line 14: This is unclear. Are then same exact oxidant fields

(climatology) used for both the CLASSIC and GLOMAP runs? If not, why? Wouldn't

this make a difference?

*The oxidant fields differ as the CLASSIC climatology had been generated separately from an earlier simulation. This is now made explicit in the text. Whilst it is true that these are not consistent the oxidants have no direct impact on the simulation of biomass burning aerosol in either CLASSIC or GLOMAP-mode. Impacts are limited to the relatively minor secondary impacts of oxidation of sulphate, which mix internally with BB aerosol. The comprehensive nature of implementing an entirely different aerosol and chemistry scheme (CLASSIC vs GLOMAP-mode) means that different oxidant fields have been adopted. It is difficult to unify these inputs without significant development and re-evaluation for a scheme (CLASSIC) that is no longer being actively developed. We now make this difference explicit in the text:*

***Page 6, Line 14:***

***"CLASSIC used a climatology of oxidants generated separately from an earlier simulation."***

6. Page 7, Line 15: This seems odd. No sea salt transport over

land? You just wash it out once it crosses the coastline? Is this realistic?

*The sea salt mass is not an advected tracer so only occurs over ocean. This point wasn't clear so I explain a little better:*

***Page 7, Line 14:***

***"CLASSIC uses a diagnostic scheme for wind-driven sea salt, i.e. sea salt aerosol is not transported but instead is diagnosed locally over ocean points as a function of wind speed and with a prescribed scale-height in the vertical (see Bellouin et al., 2011, Jones et al., 2001)."***

7. Page 7, Line 7: I'm a bit confused about how emissions are prescribed in CLASSIC. Doesn't the OC:BC ratio on emission come from the emissions inventory (GFED)? Please clarify.

*This point is now explained in more detail.:*

**Page 7, Line 27:**

**"The total aerosol mass emitted into the fresh mode is taken as the sum of BC and OC from GFED but the model makes its own assumptions regarding the proportion of BC and OC in each BBA mode. Each BBA mode is assumed to be an internal mixture of Black Carbon (BC) and Organic Carbon (OC) with an organic carbon mass fraction of 91.5 % for the fresh mode and 94.6 % for the aged mode".**

8. Page 7, Line 25 (and thereafter): Put a space between number and unit (micrometer).

9. Page 7, Line 31: "Refractive Index" -> "refractive index"

10: Page 9, Line 14: "hygroscopic" -> "hydrophilic"

***Done***

11. Page 9, Line 16: Do you get optical properties on the fly or from a pre-computed look-up table?

*Yes, we do use pre-computed look-up-tables. This is now made explicit (near end of final paragraph in section 2.2.2).*

**Page 9, Line 25:**

**"Aerosol optical properties are derived for each mode as function on aerosol mode diameter and RI using look-up-tables with pre-computed results from Mie theory. For these the RI is computed by volume-weighted averages depending on the mixture of components within any given mode."**

12. Page 9, Line 19: You do not consider UV absorption (brown carbon)? Why not? Can you indicate if this should be the case here? Does SAMBBA lend support for a representation of brown carbon?

*Neither CLASSIC or GLOMAP-mode currently have a representation of brown carbon but this as an important issue that we are considering for future developments to GLOMAP-mode. This is a challenging issue mainly the huge diversity of organics and uncertainty in measurements of brown carbon and also in how to represent this globally with only one or two tracers for organic carbon. SAMBBA did not lend any specific support for a representation of brown carbon, mainly as measurements of spectral absorption were not available from the instruments deployed in the field. The issue of brown carbon is mentioned at the end of the conclusions section.*

13. Page 9, Line 29: You imply that QFED and FEER use global scaling. They do not. QFED,for example, uses biome-specific scaling. Please reword so that is clear while these inventories do scale their emissions, the scaling is more complex then multiplying by a single number globally as you do here. Considerable effort was made by these inventories in their scaling efforts.

*Yes, this is an important distinction. I have altered this sentence to include this point and also shifted it further on in the discussion so that the difference in approach compared to the globally uniform scaling factors applied in atmospheric models is more obvious:*

***Page 10, Line 16:***

***"Note, observed AODs are also used to derive biome-specific or spatially varying scaling factors in some top-down emission estimation methods such as the Quick Fire Emission Dataset QFED (Darmenov and da Silva 2013) and the Fire Energetics and Emissions Research (FEER) (Ichoku and Elison 2014)...".***

14. Page 10, Line 1: Why did you choose to use GFED as opposed to QFED or FEER? Curious.

*For global climate modelling studies we typically use GFED as the bottom-up approach is well suited to the philosophy of Earth-System models that are built up from a detailed representation of physical processes. A top-down approach such as QFED or FEER may provide more accurate emissions for recent historical periods aiding evaluations between models and observations. However, most of our climate model experiments tend to align broadly with specifications from international MIP activities (CMIP5/6, AEROCOM, AerChemMIP) that use the GFED series. We therefore choose to evaluate the model using GFED.*

15. Page 10, Line 30: When calculating optical properties, do you use the POM or OC mass?

*Yes, as explained in section 2.3.2 the mass of organic aerosol mass is scaled up to POM for GLOMAP-mode but not for CLASSIC. The calculation of optical properties for the organic aerosol component in GLOMAP-mode is given in section 2.2.2.*

16. Page 14, Line 7: Why do you just consider Aqua?

*This was a pragmatic choice because we are aware that there was some degradation / drift in the calibration of MODIS Terra towards the later years of our study period (Polashenski et al., 2015). Also, the MIDIS collection 6, level 3 merged deep-blue and dark target product was only available from Aqua.*

**We add to section 2.6, page 15, Line 7:**

**"Terra products were not included as drift in the calibration of MODIS Terra in the later years of our observation window may have affected the retrieved AODs (Polashenski et al., 2015)."**

17. Page 14, Line 16: Are you using the Level 3 product? Is this the *best to be using? Do you do any sub-sampling of your model to only consider satellite observation times? Would sub-sampling impact your comparison? You do say this on page 16 but I would comment here as well.*

*Yes, we are using level 3 MODIS data (opening sentence of section 2.6). As we did not have the facility to sub-sample the model on observation space-time points the level 3 product is the most appropriate for our comparisons. In any case we further average the data over space and time to make our comparisons with the long-term (10-year) monthly means from the model resolution (~150km). To make this point more clearly we add the following text to the end of the paragraph, section 2.7.*

**Page 16, Line 9:**

**"For MODIS the level 3 data has been further averaged to generate 10-year (long-term) monthly mean AODs at the native resolution of the atmospheric model. These are compared against the long-term mean model values without any sub-sampling of the model data on observation space-time points. Sampling biases that may arise due to the lack of sub-sampling are discussed in section 3.1."**

18. Page 16, Line 28: You say the model AOD is calculated on clear-sky RH. What does this mean? Does your model track 2 RH (clear- and cloudy- sky)? Or do you represent RH as a sub-grid PDF? Please explain.

*The Unified Model large-scale cloud scheme does include a representation for the sub-grid PDF of total water. The scheme provides the mean relative humidity*

*calculated for the cloud-free portion of the gridbox and we use this for the aerosol hydration. The methods used to construct the PDF and calculate the mean clear-sky RH are quite complicated and so we choose not to go into this within the manuscript.*

19. Page 17, Line 28: Ilorin will also be impacted by dust aerosol; is it therefore correct to conclude that your under-representation is due to biomass burning aerosol alone? Or could your dust simulations also be off?

*Yes, dust undoubtedly affects the comparison too. We add this caveat to the text.*

**Page 19, Line 11:**

**"This again suggests an under-representation of BB emissions across West Africa during Northern hemisphere winter, although the low-bias could be partly caused by a low-bias in mineral dust aerosol from the Sahara."**

20. Page 17, Line 32: How are you attributing this to BB aerosol? Couldn't it also be anthropogenic?

*The peak in total AOD during June-Sept coincides with a peak in BBAOD and shows there is a BB contribution that comes in on top of a background of other aerosols (anthropogenic, dust, etc).*

21: Page 18: Line 15: OA =POM?

*OA is used for "Organic Aerosol". The acronym is given at the start of section 3.2 and used through the manuscript. We don't use the term POM because secondary organic aerosol also contributes to the organic aerosol mass in both models.*

22. Page 19, Line 2: I still don't understand why both CLASSIC and GLOMAP do not have the BC:OA ratio coming from the emissions inventory (since that is built into the emissions).

*The issue with CLASSIC is addressed above in point 7. As explained in the text (section 3.2, paragraph 2) GLOMAP-mode does take the OC:BC ratio from the emission inventory when adding primary aerosol emissions to the model. However, the BC:OA ratio in aerosol mass concentrations can vary because (a) there are other primary and secondary sources of carbonaceous aerosol in the model, (b) OA added from primary emissions is assumed to be 1.4 * OC emission.*

23: Page 20, Line 9: Remove space before %

*There is a single space before each "%" in this manuscript, as recommended in the manuscript preparation guidance.*

24: Page 22, Line 24: "Sampling issues (e.g. altitude) may be a large source of representativeness error in the ... "

***Suggested edit adopted (page 24, Line 19)***

25: Page 23, Line 25: Do you not also sub-sample the model at AERONET observation times? Why not? Would't that matter?

*We did not have the facility within our model configuration to sub-sample on AERONET observation times. As explained in section 2.7 we simply average over time to generate long-term monthly means. This is not ideal and we are working towards building more observation simulators into our modelling systems but this is not a trivial task.*

26: Page 24, Line 27: Comment on why it's expected that a microphysical model can achieve more realistic variability in optical properties than a mass-based model. This is an important point.

*This is a complex point. Microphysical models have the potential to achieve more realistic variability in optical properties than mass-based schemes but this isn't necessarily a given. In CLASSIC the microphysical properties were guided by in-situ measurements and this empirical approach can prove relatively accurate in representing the mean optical properties (Haywood et al., 2003; Johnson et al., 2008a).Variability in total optical properties in this approach is then dependent on external mixing between aerosol species and on the parameterized variability with relative humidity, rather than on a representation of microphysical processes. Note, the CLASSIC BBA species is itself represented optically as an internally mixed particle with specified mass fractions of BC and OC. Natural variability in optical properties can potentially be simulated much better with the more complex microphysical models. These can also represent the process of internal mixing between modes. However, these include many uncertain processes that mass-based models do not need. Therefore, we prefer not to set the expectation that one scheme is better than the other but rather assess the two approaches as objectively as possible. Overall GLOMAP-mode compares better than CLASSIC to the observed variability in optical properties as mentioned in the second sentence of the conclusions.*

27. Page 25,

Line 30: "Scattering Growth Factor" -> "scattering growth factor"

***Done***

28. Page 30, Line 8:

This is a very important point. Can you comment on why the growth factor is smaller in the observations? Do any of the chemical data suggest reasons for this? Do you think this is characteristic of all BB or just over south America?

*Section 3.5 contains a detailed discussion of the observations and modelled growth factors. There are huge discrepancies between the three observation sets (MH03, KH98, SW08) and the reasons for this are not fully understood. The models actually agree with the observations from MH03 (S. Africa) but the observed growth curves from KR98 (Brazil) are much lower. As we state, "This is difficult to reconcile, especially as both were derived from [the same] airborne humidified nephelometer system." We consider that: "Possibly the regional aerosol mixture (categorised as "regional air" in MH03) contained a substantial proportion of highly hygroscopic sulphate from industrial sources in Southern Africa and is therefore not representative of purely carbonaceous aerosol." The H-TDMA measurements reviewed more recently in SW08 suggest an intermediate level of water uptake. These include ground-based data from several Amazonian campaigns and will not suffer some of the problems associated with aerosol sampling on aircraft. In the conclusions we therefore suggest (final paragraph of section 3.5 & paragraph 3 of section 4) that the models overestimate the water uptake, and emphasize the range from the H-TDMA observations in SW08.*

29: Figure 3: Could remove repeated colorbars (since all are the same) and just have one large one.

*Done for all AOD contours plots (Figures 2 – 4).*

What is the white in the squares in (e)? Missing data? Maybe grey this out.

*Yes, missing data values from MODIS are plotted as white. I add this detail to Figure caption 2. Using white causes the areas to stand out as missing, whereas grey is harder to distinguish from the pale blue colours and could be mistaken for values around 0.05-0.1.*

30: Figure 5: For AERONET, put the standard deviation on the month to indicate sub-monthly variability Could also do this for the model as shading.

*The uncertainty in the monthly mean AODs have now been indicated in Figure 5 by plotting vertical lines showing +/- 1 standard error. Shading had been tried for this plot but made it too difficult distinguish between the 5 plotted lines.*

***In section 3.1, page 19, Line 22:***

***"The approach we have taken is to average over 10-years of data to gain more confidence in the long-term monthly means. The standard error on the monthly means AODs are generally much smaller than the differences between observed and modelled values, indicating that our results are not strongly biased by interannual variability of either the simulated or observed AOD. There main exceptions are for August at Bonanza Creek and Aug – Sept at Alta Floresta where the larger standard error in AERONET AOD indicates that interannual variability has a strong impact on the comparison."***

31: Figure 8: The part labeled "dust-dominated" -> is there another measurement that confirms this? You see the same thing (almost) in SAFARI-2000 and there was not as much dust in southern Africa correct?

*The evidence for this comes from Johnson et al. (2008a) where by sampling in and out of plumes the signature of the smoke aerosol size distribution is clearly distinct from the dusty background. By comparing samples in this way we found that the smoke contributed very little to particle numbers for d > 0.8um. In SAFARI-2000 you see the same shape to the PCASP size distribution but with the number concentrations for d > 0.8um an order of magnitude less. This could still be dust as there are some dust sources in Southern Africa (e.g. dry river beds along the Namibian coastline).*

 32. Figure 11: Again, one colorer for each 2 panels across.

*Given that the contour levels do vary between panels it is maybe clearer for each panel to include it's own colour bar.*

33. Figure 12: Rather than all the grey spaghetti lines, why not just shade max-min? Would be easier to see

*Possibly, but because there are few data points in some months and sometimes outliers, shading the min-max range gives a somewhat misleading impression. The spaghetti lines were used so that you can see the min-max range and also a feel for the number of data points each month and how they are spread.*

---

## Author Comment (AC2) · 29 Sep 2016

**Response to comments from reviewer # 2:**

**"Evaluation of biomass burning aerosols in the HadGEM3 climate model with observations from the SAMBBA field campaign"**

Reviewer comments in normal font

*Responses in italics,* **change to manuscript text shown in bold**

The authors present an evaluation of the performance of two aerosol simulation modules used within HadGEM, against a range of observations. The manuscript is thorough and well written, and falls within the scope of the journal. Some of the arguments made require some further clarification or quantification. After this and some other minor issues are dealt with, I recommend publication in ACP.

*Thanks for the positive feedback*

Main comments

My main question is about the scaling factors of GFED emissions used. I understand that they were chosen so as to "match the magnitude of AOD observations" (p16,l7) / "give good agreement between modelled and observed mid-visible AOD" (p9,l27). However, no further description of how the exact scalings were chosen is given, and yet the scaling factors make it all the way into the abstract, where they read as a conclusion about emission underestimates by GFED. Even slightly different scaling factors might change the discussion of AOD comparisons in section 3.1, in particular regional features such as discussed e.g. on page 16, lines 18-20. From the global/regional AOD values given in Figures 2-4, which reflect the entire aerosol distribution in the model, it seems that there's some range of scaling factors that might fit (as the authors also discuss on page 31. I'd suggest adding some sensitivity studies here, quantifying (through model bias calculations or similar) that any AOD comparison improvement in GLOMAP over CLASSIC is robust, and maybe toning down the focus on the scaling factors to be more in line with the authors own discussions in the Conclusion section.

Further, it's difficult to properly see from Figures 2-4 what regions are most affected by the improved aerosol treatment. I suggest adding some difference or ratio plots (maybe just versus one of the MODIS collections) to highlight the changes.

*AOD scaling issue*

*I can understand the lack of satisfaction with the global scaling factors. The need to scale the BB emissions is a problematic issue that many other modelling studies have encountered (references given in section 2.3.1). Prior to scaling the simulation of AOD in BB regions was lacking in both CLASSIC and GLOMAP-mode and so the purpose of applying the scaling factors was to highlight this issue, show the approximate magnitude of the discrepancy, and show that it varied between CLASSIC and GLOMAP-mode due to differences in the two aerosol schemes.*

*As the reviewer has surmised, the scaling factors were not calculated precisely but resulted from experimenting with different scaling factors and assessing the results across the AERONET and MODIS AOD comparisons we present. We also looked at the modelled BBAOD from CLASSIC and GLOMAP-mode to ensure these were consistent in magnitude across the main BB source regions after scaling.*

    *1) To make this clearly in the manuscript the following text has been added in section 2.3.1:*

**Page 10, Line 8:**

**"These scaling factors were not calculated precisely but were found to give good overall correspondence between modelled and observed peak AODs over continental BB source regions in the tropics, and tuned to ensure a consistent AOD contribution from BB emissions in CLASSIC and GLOMAP-mode, over the BB source regions."**

    *2) The agreement between CLASSIC and GLOMAP-mode BBAOD is explained more clearly in section 3.1:*

**Page 17, Line 6:**

**"The magnitude of BBAOD is also very similar in both models as BB aerosol emissions were scaled separately in each model to ensure the modelled AOD approximately matched MODIS and AERONET AODs observed over the main BB source regions during peak BB months where BB was the dominant contributor to modelled AOD."**

    *3) As recommended by the reviewer we have also added plots of unscaled AOD to Figures 2 – 4 to allow the reader to see the impact of scaling on total AOD. This is now discussed in section 3.1:*

**Page 17, Line 11:**

**"The impact of the BB aerosol emission scaling factors is shown by comparing the total modelled AOD from scaled and unscaled simulations. The emission scaling factors have a relatively modest impact on the global distribution of AOD when assessed on an annual mean basis. This is due to**

*the highly seasonal nature of BB emissions. Nevertheless, even in annual means, it is clear that AODs over tropical South America and Africa are somewhat lower than observed (from both MODIS collection 5 and 6) in the un-scaled simulations. The scaling factors bring modelled AOD closer to the observations, although the benefit is clearer in later figures (3 – 5)."*

*Other minor edits have been made in section 2.4 (experimental design) to include the description of the unscaled simulations. Also some minor edits in section 3.1.1 and 3.1.2 (AOD assessment) to contrast their result with the scaled result.*

*In section 3.1.2 we add:*

**Page 18, Line 11:**

**"Over the BB regions the modelled AOD is generally underestimated in the un-scaled simulations compared to MODIS. In the scaled simulations total AODs agree very well with MODIS, especially over South America and Indonesia. However, some discrepancies between modelled and observed AODs remain...".**

**Page 18, Line 23:**

 **"This leads to an overestimate of modelled AOD over central Africa in the scaled simulations."**

4) *For the benefit of the reviewers and editor, below we also include scatter plots of model versus AERONET and MOIDIS AOD below (Figs R1 & R2).*

*This demonstrates the difficulty we had in deriving an appropriate global scaling factor. Whilst it is obvious that AODs in BB-affected regions are biased low prior to scaling there is a great deal of spread. The magnitude of the discrepancy varies between different regions / AERONET sites and to some extent as a function of AOD. There may be various reasons for this spread, such as the lack of sub-sampling of the model to observations space-time points, and biases in simulated water uptake and in other aerosol sources. Even after selecting regions and months where BB emissions are the most dominant, other aerosol sources still contributed 20-50% to modelled AOD (in the scatter plots we filtered data to remove points where BBAOD contributed less than 50% to the scaled AOD). Observations may also be biased in some regions. For instance, we have low confidence in the high MODIS AODs (0.6 – 1.2) over the SE Atlantic Ocean due to low sampling statistics (data points circled in Fig R2 below). Intensive observations over the SE Atlantic via the ORACLES, CLARIFY and AEROCLO-SA will help elucidate this problem (Zuidema et al., 2016)\*. We do have reasonable confidence in the high AODs observed over Southern Amazonia and Southern Africa during Aug – Oct where AERONET (Alta Floresta and Mongu) and MODIS are in reasonable agreement. For these reasons it did not seem possible (or appropriate) to devise a precise method for deriving the global scaling factors.In the case of GLOMAP-mode the scaling*

*factor of 2.0 results in good agreement between modelled peak AODs over Amazonia and Southern Africa (Aug – Oct) and the AERONET observations and MODIS observations (Figs R1 & R2 below). In the case of CLASSIC this was scaled to the nearest 0.1 to give the best consistency between CLASSIC and GLOMAP-mode for the BBAOD and AOD in the main BB regions equalled that of GLOMAP-mode, as shown below in Fig R3, and in Figs 2 – 5 of the manuscript.*

*Zuidema, P., J. Redemann, J.M. Haywood, R. Wood, S. Piketh, M. Hipondoka, P. Formenti, Smoke and Clouds above the Southeast Atlantic: Upcoming Field Campaigns Probe Absorbing Aerosol's Impact on Climate, Bulletin of the American Meteorological Society, DOI: http://dx.doi.org/10.1175/BAMS-D-15-00082.1, 2016.*

*In the abstract we do not claim that this scaling factor is due to bias in GFED3.1. Rather we note how the scaling factor depends on assumptions within the aerosol schemes:*

*Page 1, Line 19:*

*"Finally, good agreement between observed and modelled AOD was gained only after scaling up GFED3 emissions by a factor of 1.6 for CLASSIC and 2.0 for GLOMAP-mode. We attribute this difference in scaling factor mainly to different assumptions for water uptake and the growth of aerosol mass during ageing via oxidation and condensation of organics."*

*Similarly, in the conclusions we do not put forward these scaling factors as a conclusion about the magnitude of GFED3.1 biases or as recommended scaling factors for other models to adopt. Rather we discuss the various issues and areas of uncertainties that need resolving to make sense of this scaling problem.*

5) *To stress this further the follow text has been added to paragraph 5 of the conclusions:*

**Page 33, Line 5:**

**"Moreover, due to the difficulties of comparing large-scale models with limited observations, these scaling factors are not precise but rather indicate the approximate scale of the AOD biases."**

Figure 7 is another example where I don't quite follow the authors argument that GLOMAP is a clear improvement (page 19-20). It seems that some observed species ratios are closer to CLASSIC, some to GLOMAP, and unifying MAC to 10 m2/g also seems to pull both ways. Please add some quantification of the improvement here.

*Yes, the results in Figure 7 show that the composition from the two aerosol schemes are pretty similar with no clear winner. The only notable difference is a slightly higher BC mass fraction in GLOMAP-mode, which is generally closer to the observations.*

*For instance averaging over the four locations in Figure 7 the average BC fractions are:*

| | Obs as given | Obs with MAC 10 m2/g | CLASSIC | GLOMAP-mode |
|---|---|---|---|---|
| Avg BC frac % | 9.5 | 9.2 | 6.2 | 7.9 |

*After unifying the MAC assumption the comparison still comes out in favour of the higher BC fraction in GLOMAP-mode. I agree that we can not be very confident in this comparison as there are large uncertainties in absorption and BC measurements. We therefore do not mention this result in the abstract and now describe the apparent improvement more cautiously in section 3.2:*

*Page 21, Line 23:*

*"GLOMAP-mode gives slightly higher BC mass fractions than CLASSIC and in general GLOMAP-mode BC mass fractions are closer to observed values... **Tentatively,** GLOMAP-mode therefore shows **some** improvement over CLASSIC, although it still appears to underestimate BC mass fraction..."*

*And In the conclusions:*

*Page 32, Line 9:*

*"...These variations are a challenge for the models to capture. Whilst the dry BC mass fraction and SSA in GLOMAP-mode (7-10 %; 0.85 – 0.87) are closer to the observed values than CLASSIC (5-9 %; 0.91), the modelled variability between source regions is lower than observed.*

As the authors themselves point out (p29,l17), sensitivity tests of the injection height assumptions of BB aerosols would be very useful - both for the present analysis, and for the aerosol community.  Adding 1-2 simulations here would further increase the relevance of the paper.

*Yes, we did run some earlier experiments looking at the sensitivity to vertical injection height. At present forest emissions are elevated from 0 – 3km and savannah emissions at the surface. We tried (i) injecting all BB emissions at the surface, (ii) injecting all BB emissions uniformly from the surface to 3km). The results are included in the additional plot (Fig R4) below. Scenario (i) made a huge difference to the vertical profile of aerosol over Amazonia leading to very high concentrations in the lowest few 100m of the atmosphere. This was in part due to the fact that emissions in the model do not have a diurnal cycle and so concentrations build up in the night-time. In the comparisons for the DABEX and SAFARI-2000 regions the sensitivity to injection height was less, mainly as these are downwind*

regions with elevated BB plumes and atmospheric convection has had time to mix the surface emissions upward. However, Injecting all BB emissions from 0 – 3km rather than at the surface still has some impact on these downwind regions. These sensitivity experiments were completed in an earlier configuration of HadGEM3 and have not been repeated for the GA7 configuration that we have used in this manuscript.

*The main focus of the study has been on comparing the two aerosol schemes as they are rather than experimenting with improvements. As far as we can judge here, the models perform reasonably well with the current (rather crude) injection height assumptions. Therefore, we made a decision to leave the experimentation with injection heights as a topic to re-visit and explore in a separate study. To evaluate this properly would require more observations, especially for instance taking advantage of MISR plume heights and/or CALIPSO data. An AEROCOM activity is also focussing on this very issue and we have participated in this submitting HadGEM3 results. Hopefully this will shed some light on the injection height issue.*

Minor/technical issues

- Many references are made to two as-yet unavailable manuscripts ("Darbyshire et al., in preparation").  As comparisons to SAMBBA is a main point of the paper, and topics such as data averaging are important for reproducibility, this was somewhat annoying when trying to understand the analysis presented here.

*These papers are still in preparation at the present time. The first of these (2016a) relating to the vertical distribution of aerosol is less critical and the citation has been removed. Marenco et al. (2016) is already cited here (section 2.3.4, paragraph 1) and provides evidence of the vertical distributions of aerosol during SAMBBA from both the lidar and insitu measurements. The second (now the only) reference to ("Darbyshire et al., in preparation 2016") relates to a study summarizing the physical and optical properties of aerosols measured during SAMBBA. I understand it is not ideal for the reviewer and reader but it is planned that this paper will be submitted to ACP this autumn. This will contain detailed descriptions of the flight patterns, instrumentation and data processing methods. Only a summary can realistically be given in the present manuscript but this has been bolstered somewhat in section 2.5, paragraph 1 to explain the basic approach to data selection and averaging:*

***Page 13, Line 26:***

***"The regional averages of aerosol particle size distribution, composition and optical properties are based on data from straight level runs sampling the regional haze. Data sections corresponding to plume penetrations (identified from spikes in CO, CO2, BC and aerosol scattering) were filtered out prior to averaging."***

*As so often in dedicated observational campaigns, peer reviewed papers cannot be always be prepared / submitted in series as this would lead to undue delays in the duration for open submissions to the journal.*

- P5l21:  Please give a brief explanation of the MetUM, even if it's probably known to most readers.

*A little more detail has been added at the start of section 2.1.*

**Page 5, Line 21:**

**"This work uses global simulations of the Met Office Unified Model (MetUM), a state-of-the-art atmospheric general circulation model with a range of configurations for numerical weather prediction and climate simulations. Here we use the MetUM within the framework of the third generation Hadley Centre Environment Model (HadGEM3) (Hewitt et al., 2011)."**

- P6l20:  So SO4 and BC emissions are climatologies, but BB emissions have annual variations? Or do they change through interpolations? Please clarify.

*For both anthropogenic and biomass burning emissions we take the mean emission from the period 2002 – 2011 and run without any year to year change in emissions. The BB emissions do contain a seasonal cycle of course so we use monthly varying emissions. To make this clearer in section 2.1. We add, when referring to the GFED emission.:*

**Page 6, Line 17:**

**"We use monthly mean emissions averaged over the period 2002-2011."**

*And in reference to the anthropogenic emissions:*

**Page 6, Line 24:**

**"However, we keep annual emissions constant at the 2002 - 2011 mean rate."**

- P7l15: Will the assumption of no sea salt transport over land in CLASSIC affect AOD comparisons in BB-emission coastal regions?  Does this contribute to differences to GLOMAP?

*Yes, but the contribution of sea salt to AOD in the major BB regions is very small. The main aerosol sources that affect the comparison are anthropogenic emissions, dust (mainly West Africa), and biogenic (especially Amazonia and the Congo basin). These are mentioned in the discussion of results.*

- P12l1: Add "of" between "prevalence" and "moist".

**Done**

[Figure]

***Fig R1.*** *Model versus AERONET AOD. Observations are the long-term monthly mean averages shown in Figure 5 of the manuscript, except here data points are selected only if CLASSIC or GLOMAP-mode showed that at least 50% of the AOD was associated with BB emissions (i.e. BBAOD / AOD > 0.5) in the scaled simulations. The linear fit is a straight-forward linear regression (blue dotted line). Model results are shown with and without the global scaling of BB emissions.*

[Figure]

***Fig R2.*** *Model versus MODIS C6 AOD. Observations and model data points are the long-term monthly mean averages for September as used in Figure 3 of the manuscript, except here data points are shown only if CLASSIC or GLOMAP-mode showed that at least 50% of the AOD was associated with BB emissions (i.e. BBAOD / AOD > 0.5) in the scaled simulations. The linear fit is a straight-forward linear regression (blue dotted line). Model results are shown with and without the global scaling of BB emissions. Circled are data points over SE Atlantic ocean, which may be overestimated by MODIS.*

[Figure]

**Fig R3.** *CLASSIC versus GLOMAP-mode BBAOD (left) and AOD (right) for the scaled simulations. Model data points are the long-term monthly mean averages for September as used in Figure 3 of the manuscript, except here data points are shown only if CLASSIC or GLOMAP-mode showed that at least 50% of the AOD was associated with BB emissions (i.e. BBAOD / AOD > 0.5) in the scaled simulations. The linear fits are straight-forward linear regressions (blue dotted line). The plot demonstrates that the scaling factor of 1.25 applied to the BB emissions in the CLASSIC simulation leads to good agreement with the BBAOD and AOD from the scaled GLOMAP-mode simulation.*

[Figure]

**Fig R4.** *Vertical profiles of fine-mode aerosol mass concentration for the SAMBBA, DABEX and SAFARI-2000 regions (as in Figure 14 of manuscript). All lines are from earlier simulations with GLOMAP-mode based on GA6 (the previous Global Atmosphere configuration compared to simulations used in the manuscript). The light blue indicates a simulation using the default vertical injection assumption, red indicates a simulation where all BB emissions are injected uniformly from 0 – 3km, and dark purple indicates a simulation with all BB emissions injected at the surface.*

---

## Author Response (AR2)

**Response to co-editor comments on:**

**"Evaluation of biomass burning aerosols in the HadGEM3 climate model with observations from the SAMBBA field campaign"**

Thank you for your helpful comments.

We agree that there is considerable variability in the OC:OA ratio in observations and that the current ratio of 1.4 in GLOMAP-mode is at the very minimum of the observed range. Increasing this would have an important bearing on the global emission scaling factors and so this is an important point to bring out more strongly in the manuscript.

From reviewing the literature estimates of OA:OC range from 1.4 to 3.2 for organic aerosol mixtures, but with a clear trend towards increasing OA:OC with more aged aerosol mixtures. This makes it is difficult to determine an upper bound for POM:OC in the freshly emitted particles. In section 2.3.2 we'd concluded:

"*Recent analyses of aerosol mass spectra (e.g. Aitken et al., 2008; Ng et al., 2010; Tiitta et al., 2014; Brito et al., 2014) and preliminary analysis of airborne data from SAMBBA indicate POM:OC ratios in the range 1.5 – 1.8 for fresh particles / near-source emissions. However, the observations indicate considerable variability with aerosol age and source region with POM:OC ratios increasing to 2.0-2.3 for aged and more highly oxidised aerosol. This introduces considerable uncertainty in gauging a representative POM:OC for global models where near-source ageing may not be represented.*"

There is also some uncertainty for global models on whether a value representative of freshly emitted particles is relevant or a value representing "young" aerosol after some near-source ageing. This depends on whether there is a representation for near-source ageing in the model (there is not yet for GLOMAP-mode). Therefore, there is certainly scope to increase OC:POM in GLOMAP-mode and decrease the global emission scaling factor. As an example, the global emission scaling factor for GLOMAP-mode could be reduced to 1.5 if the POM:OC ratio was increased to 1.9, which would still be a credible value for BB emissions. Doing this would reduce the BC:OA ratio and perhaps require revisions to BC refractive index or OC refractive index (brown carbon) to compensate.

Please find the revised manuscript with tracked changes in red and comments to explain how the change addresses the comments. The text in the manuscript has been altered in the following three sections to bring out these points:

(i) Abstract (ii) Section 2.3.1 (Global emission scaling factor (iii) Section 4 (Conclusions)

**Evaluation of biomass burning aerosols in the HadGEM3 climate model with observations from the SAMBBA field campaign**

B. T. Johnson[1], J. M. Haywood[1,2], J. M. Langridge[1], E. Darbyshire[3], W. T. Morgan[3], K. Szpek[1], J. Brooke[1], F. Marenco[1], H. Coe[3], P. Artaxo[4], K. M. Longo[5*], J. Mulcahy[1], G. Mann[6], M. Dalvi[1], and N. Bellouin[7]

[1]Met Office, Exeter, UK

[2]CEMPS, University of Exeter, Exeter, UK

[3]Centre for Atmospheric Science, University of Manchester, Manchester, UK

[4]Physics Institute, University of São Paulo, São Paulo, Brazil

[5]National Institute for Space Research (INPE), São José dos Campos, Brazil

*Now at NASA Goddard Space Flight Center and USRA/GESTAR, Greenbelt, MD, USA

[6]National Centre for Atmospheric Science, School of Earth and Environment, University of Leeds, Leeds, UK

[7]Department of Meteorology, University of Reading, Reading, UK

Correspondence email: ben.johnson@metoffice.gov.uk

**Abstract**

We present observations of biomass burning aerosol from the South American Biomass Burning Analysis (SAMBBA) and other measurement campaigns, and use these to evaluate the representation of biomass burning aerosol properties and processes in a state-of-the-art climate model. The evaluation includes detailed comparisons with aircraft and ground data, along with remote sensing observations from MODIS and AERONET. We demonstrate several improvements to aerosol properties following the implementation of the GLOMAP-mode modal aerosol scheme in the HadGEM3 climate model. This predicts the particle size distribution, composition and optical properties, giving increased accuracy in the representation of aerosol properties and physical-chemical processes over the CLASSIC bulk aerosol scheme previously used in HadGEM2. Although both models give similar regional distributions of carbonaceous aerosol mass and Aerosol Optical Depth (AOD), GLOMAP-mode is better able to capture the observed size distribution, single scattering albedo, and Ångström exponent across different tropical biomass burning source regions. Both aerosol schemes overestimate the uptake of water compared to recent observations, CLASSIC more so than GLOMAP-mode, leading to a likely overestimation of aerosol scattering, AOD and single scattering albedo at high relative humidity. Observed aerosol vertical distributions were well captured when biomass burning aerosol emissions were injected uniformly from the surface to 3 km. Finally, good agreement between observed and modelled AOD was gained only after scaling up GFED3 emissions by a factor of 1.6 for CLASSIC and 2.0 for GLOMAP-mode. We attribute this difference in scaling factor mainly to different assumptions for the water uptake and growth of aerosol mass during ageing via oxidation and condensation of organics. We also note that similar agreement with observed AOD could have achieved with lower scaling factors if the ratio of organic carbon to primary organic matter was increased in the models toward the upper range of observed values. Improved knowledge from measurements is required to reduced uncertainties in emission ratios for black carbon and organic carbon, and the ratio of organic carbon to primary organic matter for primary emissions from biomass burning.

**Comment [b1]:** As suggested by the co-editor we bring out the issue with OC:POM ratio and emission ratios in the abstract.

[revised manuscript text omitted]

**Comment [b3]:** Some more substantial discussion here of how OC:POM ratios could be increased and the impacts this would have on BC mass, and absorption / SSA. This brings out the final points made in the co-editor's comments and we hope fully addresses the issue.

Note, we did not feel that the emission scaling factors could be completely as recent observations of OC:POM ratio in fresh BB plumes (refs in the text) generally have values less than 2. Higher OC:POM ratios are certainly observed in aged mixtures but we would suggest that models need to represent the ageing processes that lead to this enhanced mass.

[revised manuscript text omitted]